



# Analyses of Peace River SWIPS data and its implications for the roles played by frazil ice and *in situ* anchor ice growth in a freezing river

**John R. Marko[1] and David R. Topham[1]**

[1]ASL Environmental Sciences Inc., Saanichton, BC, Canada

**Correspondence:** John R. Marko (jmarko@aslenv.com)

**Abstract**

Peace River SWIPS (Shallow Water Ice Profiling Sonar) data were analyzed to assess and quantify the roles of frazil ice suspensions and riverbed anchor ice grown *in situ* during the initial buildup of a seasonal ice cover. Data were derived from quasi-continuous monitoring of frazil parameters throughout the water
column to provide direct and indirect measures of anchor ice volume and mass growth rates. Analyses utilized water level and air and water temperature information and directly measured acoustic volume backscattering coefficients to track and interpret spatial and temporal changes in riverbed and water column ice constituents. Interests were focused on 4 specific frazil intervals characterized by anomalously low levels of frazil content (relative to simulations with an anchor ice-free river ice model) distinguished by two
strikingly different types of time dependences. A simple physical model was proposed to quantitatively account for discrepancies between measured and simulated results in terms of the pronounced dominance of anchor ice as an initial source of river ice volume and mass. The distinctive differences in temporally variable water column frazil content are attributed, in this model, to corresponding differences in the stabilities of riverbed anchor ice layers against detachment and buoyancy-driven movement to the river
surface. In accord with earlier observations, the stability of *in situ* grown riverbed ice layers appears to be inversely proportional to cooling rates. The strength of the coupling between the two studied ice species was shown to be strong enough to detect changes in the anchor ice constituent from variations in water column frazil content.

## 1. Introduction

SWIPS (Shallow Water Ice Profiling Sonar) results obtained in early studies of Peace River freeze-up periods (Jasek et al., 2005; Marko and Jasek, 2010a,b) were indicative of highly dynamic surface- and frazil-ice environments. More ambiguous, but definitive, observations of physical instabilities in deployed instruments as well as blockages of the upward-looking SWIPS acoustic beams suggested substantial amounts of anchor ice were also episodically present. This ice was presumed to have been produced
(Hammar and Shen, 1995) by adhesion of mobile frazil particles on instrument surfaces. Nevertheless, measurement results obtained by Jasek et al. (2011), in conjunction with simulations with the widely used CRISSP1D river ice model (Shen, 2005) suggested that anchor ice was not a major factor in ice cover development. Instead, simulations showed timings and magnitudes of observed surface ice changes to be explicable in terms of buoyancy-driven surfacing of water column frazil.

This conclusion was seriously called into question when initial reports on 2011-2012 SWIPS Peace River measurements (Jasek et al. 2013; Marko et al., 2015) showed frazil fractional volumes rarely rising as high as 0.01% during supercooling periods. Typical time-dependent frazil fractional volumes, $F(t)$, over the bulk of individual intervals, approximately 0.002%, were two orders of magnitude below the 0.3% levels which simulations suggested could be sustained during an extended frazil event. The latter, anticipated, fractional
volumes were typical of frazil growth in laboratory tanks and flumes (Ettema et al., 1984; Ettema et al., 2003; Ye et al., 2004) and, in the absence of contrary field data, were assumed to be attainable in river



settings. It was suggested by Marko et al. (2015) that the anomalously low measured frazil contents were consequences of ice model neglect of anchor ice growth which suppressed frazil production.

Possibly reflecting the magnitude of the reported deviations, this conclusion and the underlying SWIPS results were received with some skepticism. In particular, objections were raised about the absence of SWIPS instrument calibrations on frazil targets in spite of the difficulties known to be associated with such a calibration approach (Ghobrial et al., 2013; Marko and Topham, 2017). These objections also ignored the well-established successes of acoustic profiling and the surrogate calibration procedures utilized both in connection with the Peace River work (Marko and Topham, 2015) and in previous applications to suspensions of sediments, zooplankton and other target materials. Surrogate approaches, in particular, have rarely been questioned since key calibration issues, such as sensitivities to acoustic frequency and target shape are often either material-independent or simply linked to known mass density and sound speed parameters. It can be argued that surrogate testing is particularly suited for frazil applications in which target stability and control are not easily attained.

The potential shortcomings of earlier calibrations have now been addressed in the foregoing paper (Topham and Marko, 2020) by further analyses of in-hand surrogate results and consistency tests on river frazil acoustic data. These analyses delved into the details of the extraction processes applied to data acquired in simultaneously both two- and three-different acoustic frequency channels. Particular efforts were given to assessing errors introduced by applying a theory of scattering by spherical targets to, primarily, disk-shaped frazil particles. Fractional volume was found to be a robust descriptive parameter for fundamental reasons intrinsic to elastic scattering theories. The multifrequency approach facilitated refinements in the assumed dependences of cross sections on $k_1a_e$, the critical product of wavenumber and particle effective radius which was the key scaling parameter in the utilized Faran Effective Sphere Theory (FEST). These adjustments corresponded to a small, 25%, increase relative to earlier fully FEST-based estimates: yielding accuracies largely limited by +/-30% systematic uncertainties inherent to the underlying acoustic transceiver calibrations. In short, the new validations strongly support the reality of major deviations of the Marko et al. (2015) frazil estimates from expectations based upon model simulations which neglect anchor ice growth.

This situation is not inconsistent with recent recognitions (Kalke et al., 2015; Kempema et al., 2015; Evans et al., 2017); McFarlane et al., 2017) that riverbed-grown anchor ice is a common constituent of seasonal ice covers. Nevertheless, the absence of quantitative data on both riverbed ice and water column frazil content greatly inhibit monitoring and modelling improvements such as those recently suggested by Makkonen and Tikanmati (2018). The simplest remedy for this absence and consequent uncertainties in the anchor ice/frazil ice relationship would be to combine quasi-continuous SWIPS measurements of $F(t)$ throughout the water column with contemporary river and environmental data collection. Quantitative estimates of anchor ice growth rates can be derived from such data with simple thermodynamic balance calculations (Osterkamp, 1978). Given the very low frazil fractional volumes reported by Marko et al. (2015) and the quoted uncertainties, such calculations could be expected to provide anchor ice growth data with precisions more than sufficient to support model refinements. The scope and importance of river ice growth issues suggest that it is long past time to begin to utilize the quantitative outputs of SWIPS measurements to address persisting major gaps in understanding ice cover development. The present work undertakes this task by using results from the early winter portions of the 2011-2012 Peace River field program to derive and test the effectiveness of a simple physical model of the initial stages of the development process which is based upon widespread *in situ* growth of anchor ice during frazil events.

Our treatment begins in Sect. 2 with brief descriptions of: the Peace River study region, the deployed instrumentation and the CRISSP1D model. This Section also includes a concise summary of the main





features of the multifrequency SWIPS measurement and data processing approach as applied to the Peace River data sets. The analytical and interpretative development in Sect. 3 conveniently segments pre-consolidation frazil events according to differences in the observed time dependences of the key frazil fractional volume parameter, $F(t)$. A variety of data and observations relevant to interpreting two alternative

types of frazil events provides a basis for formulating basic models of anchor and frazil ice behaviour during such events. Model components and interpretative conclusions are summarized in a final Sect.4 prior to discussions of implications for efforts to improve river simulations from results obtained in productive field and laboratory measurement programs.

## 2. Deployment, instrumentation, data extraction and comparison methodologies

### 2.1 Deployment and instrumentation

Data were acquired between November, 2011 and April, 2012 by BC Hydro at a monitoring site on the Peace River near Town of Peace River (TPR), Alberta. Measurements utilized a weighted, electrically heated, instrument package deployed on the riverbed (Fig.1) in 5 m to 6 m of water 25 m off the River's south bank. Armoured power, control and data acquisition cables linked the submerged instruments to a

shore station. Acoustic frazil profile measurements utilized a 4 frequency Shallow Water Ice Profiling Sonar (SWIPS) unit (manufactured by ASL Environmental Sciences Inc.) operating at 125 kHz, 235 kHz, 455 kHz and 774 kHz with upward-looking, acoustic transmitting/receiving transducers (transceivers). The instrument was calibrated to accurately measure volume backscattering coefficients, $s_v$, in each channel. These coefficients denote the fractions of acoustic power incident upon a unit volume of diffusely

suspended targets which is scattered directly back toward the power source. The transducers for channels 1 (125 kHz), 3 (455 kHz) and 4 (774 kHz)) were mounted in a common moulded head attached to a pressure case separate from, but connected to, a second pressure case containing the instrument electronics and the isolated (by 30 cm) channel 2 (235 kHz) transceiver. Additional instrument and deployment details are available in Marko et al. (2015) and Topham and Marko (2020) (accompanying paper). The SWIPS

instrument was similar to the AZFP (Acoustic Zooplankton Fish Profiler) used in the Marko and Topham (2015) laboratory calibrations which included additional logarithmic signal detection capabilities and mounting of all transducers in a common head. Unfortunately, data collected with the isolated SWIPS 235 kHz channel exhibited problematic instabilities (Marko et al., 2015) which precluded use in our analyses.

Individual acoustic pulses were transmitted and received at 1 Hz in each channel. Averaging over two

adjacent time samples of return voltage provided measures of backscattering from successive 4 cm range cells. Water temperature, hydrostatic pressure, flow speed and direction profile data were acquired on a Teledyne RDI Sentinel Acoustic Doppler Current Profiler (ADCP) in the instrument package.

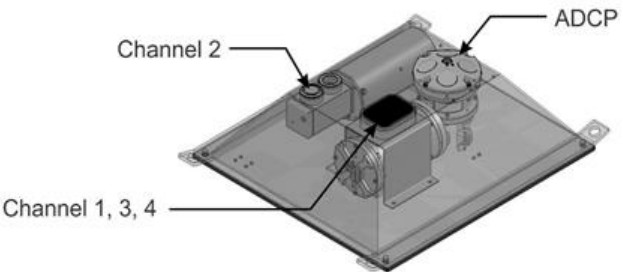

**Figure 1**. The deployed instrument package showing the locations of the multifrequency SWIPS and ADCP

current profiler including the locations of the SWIPS transducers.





### 2.2 SWIPS measurement, data extraction and model comparison methodologies

The analysis framework used in this work assumes availability of volume backscattering coefficient time series data, $s_V$, at, at least, three different acoustic frequencies. Frazil characterizations utilized data averaged over contiguous 2 minute time intervals during seven separate events spanning roughly 2.5% of the 5 month SWIPS monitoring period. Detailed analyses were confined to four of five major supercooling events, listed in Table 1, which preceded seasonal Ice cover consolidation at TPR.

**Table 1.** Analyzed frazil intervals

| Interval | Start and end times, dates | Duration (hrs) |
|---|---|---|
| 1 | 17:14 Nov. 20 to 07:00 Nov 21 | 14 |
| 2 | 01:34 Jan.3 to 12:55 Jan. | 11 |
| 3 | 19:34 Jan 14 to 23:57 Jan 15 | 29 |
| 4 | 07:34 Jan.25 to 11:59 Jan.26 | 29 |
| 5 | 23:04 Feb 6 to 08:00 Feb. 7 | 9 |

Given the heavy reliance of river ice models on frazil ice volume, it was essential that the SWIPS-measured quantities, $s_V$, be easily linked to volume-related parameters. This required access to a valid theoretical expression for $s_V$ expressible in terms of linear dimensions unambiguously convertible into scattering target volumes. In general terms, the required relationship can be written as:

$$s_V^{Theo}(v_i) = N \int_0^\infty g(a_e)\,\sigma_{BS}(a_e, v_i)\, da_e \qquad , \qquad (1)$$

where N denotes the number of particles per unit volume; $\sigma_{BS}(a_e, v_i)$ is the theoretical backscattering cross section of a spherical particle with an "effective radius" $a_e$ at an acoustic frequency $v_i$ and $g(a_e)$ defines a probability distribution normalized to unity and expressed in a two parameter lognormal form:

$$g(a_e, a_m, b) = \left[(2\pi)^{0.5} b a_e\right]^{-1} e^{-0.5\left(\frac{\ln(a_e/a_m)}{b}\right)^2} \qquad . \qquad (2)$$

The effective radius (Ashton, 1982) is defined as the radius of an "effective sphere" having a volume equal to that of the represented particle. The two population parameters in Eq. 2, $a_m$ and $b$, are, respectively, the mean value of the effective radius and the standard deviation of the logarithm of $a_e$ which specifies the spread in latter parameter. Given access to $s_V$ data at, at least, three frequencies, population descriptions in terms of $N$, $a_m$ and $b$ are obtained by minimizing a residual quantity, $q$, defined as the sum over all channels of squared differences between measured and theoretical logarithmic backscattering coefficients $S_V \equiv 10 \log(s_V)$:

$$q = \sum_{i=1}^{i=3} \left[S_V^{meas}(v_i) - S_V^{Theo}(v_i)\right]^2 \qquad . \qquad (3)$$

The optimal population parameters allow fractional ice volumes, $F$, to be calculated from:



$$F = N \int_0^\infty \left(\frac{4\pi}{3}\right) a_e^3 g(a_e, a_m, b)\, da_e \qquad . \tag{4}$$

Valid measurements and extraction of frazil information requires the applicability of Eq. 1 and confidence in the cross section relationship, $\sigma_{BS}(a_e, v_i)$. The Marko and Topham (2015) testing established the validity of linear dependences on numerical target concentrations to values of N up to and often above $10^7 m^{-3}$. The FEST cross sections, $\sigma_{BS}(a_e, v_i)$, can be expressed compactly in terms of modal series coefficients, $\eta_m$, (Stanton et al., 1989) as:

$$\sigma_{BS}(a_e, v_i) = \left(\frac{\pi a_e^2}{4}\right)\left[\left(\frac{i}{k_i}\right)\sum_{m=0}^\infty \eta_m\, (-1)^m\right]^2 \qquad . \tag{5}$$

These cross sections were calculated using the formulation developed by Dezhang Chu of the Northwest Fisheries Center. In logarithmic terms the formulation's accuracy was estimated to be better than 0.001 dB. Target-shape related deviations from FEST cross sections, which appear for $k_1 a_e$ values > 0.45, required an additional corrective step suggested by the Topham and Marko (2020) validations to assure attaining $F(t)$ estimates at the +/- 30% levels achieved in transducer calibrations.

Conversion of SWIPS $s_v$ outputs in three channels into frazil population parameters was carried out with ASL Environmental Sciences' RUNSWIPS software based upon FEST representations of the relationship between backscattering cross sections and the key acoustic frequency and frazil effective radius parameters. Time series of $F$, $N$, $a_m$, $b$ and $q$ values for each successive averaging interval are extracted and displayed as functions of time at multiple, user-specified, ranges (or heights in the water column) relative to the common transducer surface plane. Although the variations in all frazil parameters provide important information relevant to suspension dynamics, the principal interests of the present work are in the magnitudes and time dependences of $F(t)$ at mid-water ranges. Detailed studies by (Marko et al., 2015; Topham and Marko, 2020) suggest that the values obtained for this parameter are robust and, when multiplied by a factor of 1.25, to account for higher order, shape-related, refinements of the FEST cross sections relationship, yields suitably accurate $F(t)$ estimates for comparisons with the water column-averaged frazil fractional volumes simulated by ice models.

An essential component of the study involved interpretation of Echogram plots of raw backscattered digital voltage signals (in counts) as recorded for returns received from successive emitted acoustic pulses. The colour-coded strengths of these returns were displayed by ASL's ProfileView software as functions of the ranges of the scattering targets relative to the transducer faces.

Comparisons with simulations were based upon runs of a one-dimensional CRISSP1D model (Shen, 2005) within BC Hydro's operational mode (Jasek et al., 2011) without further adjustments. These runs utilized inputs of water temperature and discharge data, available at hourly intervals from a hydroelectric site approximately 370 km upstream of the SWIPS instrument. Hourly surface air temperature inputs were obtained near river gauges 7 km and 100 km downstream of the SWIPS location as well as 94, 226, 278 and 362 km upstream. The model utilized a 30-minute time step and linear interpolations of hourly input data to produce corresponding water column-averaged $F(t)$ values. Detailed discussions of simulation procedures are available in (Jasek et al., 2011).





### 3. Observations and measurements of frazil events preceding February ice consolidation at TPR

### 3.1 Outline of analytical approach.

Five separate intervals of supercooling and frazil growth were identified in the period (Table 1) separating the first appearances of frazil ice in late November, 2011 from the mid-February consolidation of the ice cover at the TPR monitoring site. The ice conditions which evolved during these Intervals were reviewed for insights into the processes underlying seasonal ice cover development. Significant differences were apparent in the magnitudes and time dependences of the extracted frazil fractional volume parameters and in their correlations with blockages of acoustic beams. The blockage phenomenon and consequent gradual losses of frazil monitoring capability were interpreted as unmistakable evidence of anchor ice accumulation on or near exposed transducer faces. Understanding the underlying growth processes required combining basic acoustic data and observations with external information (river and atmospheric environmental data) to construct a self-consistent interpretative model of frazil events.

In doing this, it was convenient to focus on the four most intense and persistent frazil events observed prior to local ice cover consolidation. These events were further divided into 2 categories, primarily on the basis of readily apparent differences in the time-dependences of the deduced parameter, $F(t)$. The simpler generic form consisted of a single large initial peak followed by a sharp drop to lower levels of frazil content which persisted, with small variations, for the remainder of event duration. Such behaviour was observed in the March, 2012 frazil interval employed in Topham and Marko (2020) SWIPS verifications. The two studied frazil intervals of this type were closely spaced (in time) and dominated the late January-early February period which preceded local ice cover consolidation. Although not, necessarily, associated with all intervals of this type[1], both considered single-peaked intervals were terminated by anchor ice blockage. A second pair of intervals, corresponding to an alternative "multi-peaked" form was encountered in late-November and mid-January during the coldest portions of the pre-consolidation period. As indicated by the nomenclature, corresponding $F(t)$ data featured multiple sharp peaks separated by periods of low or intermediate frazil content. In contrast with single peak events, these Intervals were terminated by disappearances of frazil in the water column without evidence of acoustic beam blockage. Data on the two distinct types of events are separately analyzed in Sects. 3.2.1 and 3.2.2 to establish the characteristics of each behavioural variant as a basis for understanding the origins of the observed differences.

### 3.2.1 Single peak frazil growth Intervals

Frazil fractional volumes during the 07:34 Jan.25 to 11:59 Jan.26 (Interval 4) and 23:04 Feb. 6 to 08:00 Feb. 7) (Interval 5) intervals are plotted in Figs. 2a,b based upon 2-minute averaged $s_V$ data acquired in SWIPS channels 1, 3 and 4 from frazil 2.3 m above the transducer faces. The plotted results exhibit the classic single peak form with blockages, detectable as extended, moderately steep, decreases in $F(t)$ at the end of each Interval. The timings and intensities of the blockages, interpretable as two separate obstruction events, are summarized in Fig. 3 in terms of coarse severities at each acoustic frequency. Three severity categories are delineated: unblocked, partially blocked and completely blocked. Partial blockages occurred during transitions between the unblocked (unobstructed) and completely blocked (no returns from the water column or above) extremes. As well, in both events, blockage progress included partial daytime clearances prior to resumption of the approach to the full blockages achieved in evening or early morning hours.

---

[1] Two examples of unblocked single peak intervals were: the March 20, 2012 event studied by Topham and Marko (2020) and a short (6 hour) Jan. 3, 2011 interval (Interval 2 in Table 1). These intervals were previously documented as Intervals 2 and 6 Marko et al. (2015). A seventh, March 22, 2012, spring single peak interval in the latter study was terminated by acoustic blockage.



The first onsets of the two blockages, occurred 15,0 and 6.0 hours, respectively, after the initial appearances of water column frazil which always preceded blockage occurrences. Except for a three day February period separating the two events, data included in Fig. 3 showed mean daily temperatures remaining close to or below roughly, -5°C for times as late as Feb. 19 when the consolidated ice edge was established 3 km upstream of the SWIPS. Apart from the anomalous persistence of problematic channel 2 returns in the February Interval, blockages were both first detectable and complete in the highest frequency channels 3 and 4 but eventually encompassed all channels. Near-simultaneous clearances of the Interval 4 blockages on Feb. 3 were followed by the additional sequences of frazil formation and varying and, ultimately, complete blockage associated with frazil Interval 5 (Fig. 2b). Again, in the latter case, blockages in all channels ended near-simultaneously on Feb. 20.

A representative example of the changes in backscattering returns which allow tracking of frazil content and acoustic blockage development is displayed in Fig.4. The Echogram format utilized in the Figure provides false-colour mappings of 16-bit digital voltages representing acoustic returns in a given frequency channel as a function of time and range above SWIPS transducer faces located 29 cm above the riverbed. The plotted data, corresponding to 774 kHz (channel 4) returns, document scattering strength as a function of position in the water column during a key segment of frazil Interval 4. The depicted period included progressive extinctions of returns from water column frazil and, eventually, from much stronger surface ice and atmospheric interface targets. Annotations denote Echogram features relevant to interpreting changes during a period which began 7 hours after the initial appearance of frazil in the Interval. $F(t)$ data for the full Interval as a whole are plotted in Fig. 2a. Particular note is made in Fig.4 of the narrow strip of "close-in" returns at the lowest end of the range scale. At times prior to 08:00, Jan 26, the very high digital count values in this feature were representative of transducer "ringing" which persists, for a brief period following emission of each sound pulse. Subsequent progressive thickening of this strip was due to the buildup of anchor ice on and/or just above the transducer faces. The first evidence of such ice was apparent at, approximately, 08:00 as a slight rise in the upper boundary of this feature. The effects of such accumulations were more apparent, at approximately 10:00, Jan. 26, in the onset of very visible weakening in the strength of returns from water column frazil. Smaller concurrent reductions were apparent in the strengths of the longest range components of the saturated surface returns. Blockage impacts and the width of the close-in zone began to decrease again at, roughly, 12:00. This decrease eventually allowed recovery of water column frazil returns until 18:00 when strong growth again widened the close-in regime, eliminating water column returns and rapidly eroding river surface signals. No surface or water column returns were detectable by early morning Jan. 27. Similar sequences were observed in connection with frazil Interval 5 in which, as noted above, initial signs of acoustic blockage followed frazil onset by 6.0 h.

A common pattern was noted in which changes in the strengths of returns from water column frazil and surface targets were closely correlated with and opposed in sign to contemporary variations in the spatial extent of strong close-in returns. This pattern was fully consistent with anchor ice presence adjacent to the transducer faces in volumes large enough to partially and, later, to completely block detection of acoustic returns from water column and surface targets. A second, more practical, inference was that such sudden appearances of anchor ice above a heated surface, roughly, 30 cm above the riverbed, imposed an additional, potentially useful, constraint on river ice models. Specifically, successful models, necessarily, had to incorporate frazil/anchor ice relationships capable of accounting for these and other observations made in the water column and near the riverbed during the 2011-2012 studies.





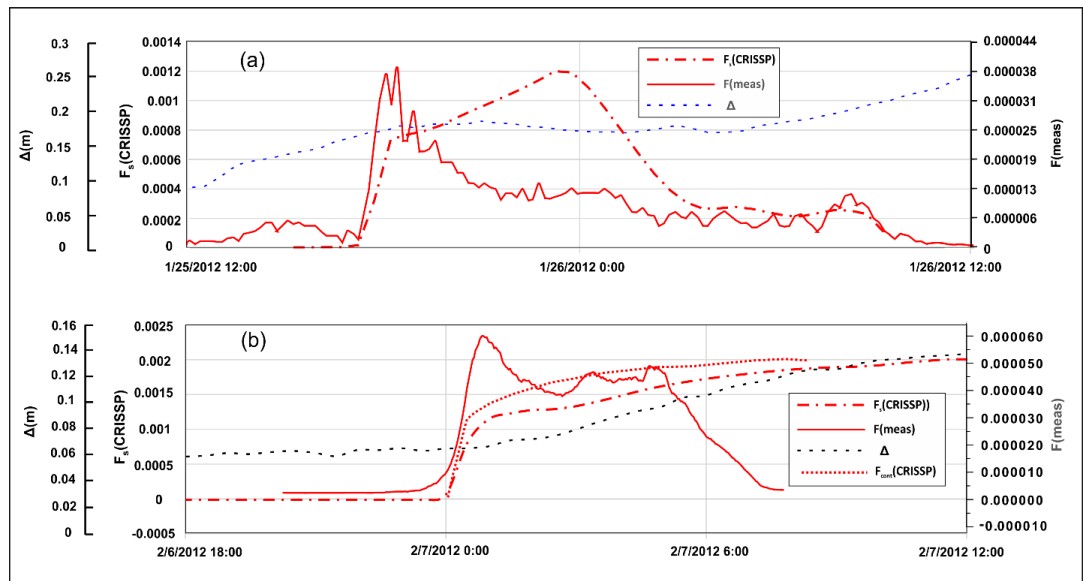

**Figure 2a,b.** Comparisons of fractional volumes as measured ($F$(meas)) and simulated ($F_s$(CRISSP), $F_{cont}$(CRISSP)) for a) Interval 4 and b) Interval 5. The $F_s$(CRISSP) simulations were shifted to earlier and later times by 11.0 and 3.5 h, respectively, to facilitate coincidence with observed frazil onsets. The plot in b) includes $F_{cont}$(CRISSP) depicting an artificially triggered simulation described in the text which uses fully contemporary post-onset environmental inputs. The measured and simulated fractional volumes were representative of, respectively measurements 2.3 m above the SWIPS instrument and water column mean values. An additional, dotted, curve represents Δ, the difference between 10-point-avergared measured water levels and those simulated by CRISSPID without allowances for anchor ice.

As suggested above, immediate insights into such relationships can be obtained by using the Osterkamp (1978) thermodynamic approach much as previously employed by Ye et al. (2004) for estimating frazil fractional volumes in a laboratory flume. In our case, it was appropriate to initially focus on the transition from frazil-free to frazil-infested conditions: assuming, at onset, a rough continuity in the rate of sensible heat loss to the atmosphere. This assumption was consistent with the negligible, few millidegree, initial warmings of the supercooled water column during early stages of frazil crystallization. In preserving continuity, energy balance required that the initial rates of frazil latent heat production (derived from immediately post-onset $F(t)$ data) had to be at least equal to pre-onset sensible heat fluxes. This equality was tested against the SWIPS-estimated rates of $F(t)$ increase plotted in Figs. 2a,b and in the equivalent multi-peaked Interval results of Figs. 5a,b. Pre-transition sensible heat fluxes, on the other hand, were calculated from the time rates of change in water temperatures measured on the ADCP instrument. These comparisons are made in Table 2 for both single- and multi-peaked intervals. Results for Interval 4 were not included in the tabulation due to notable anomalies in corresponding pre-onset water temperature data which precluded reliable estimates of rates of sensible heat loss. It can be seen that the ratios of post- to pre-onset fluxes for the three evaluated Intervals were consistently an order of magnitude or more below unity. This result suggested that most of the heat lost to the atmosphere after frazil onset did not originate from frazil growth in the water column as assumed in anchor ice-free CRISSP1D simulations (Shen, 2005; Jasek et al., 2011; Marko et al., 2015).

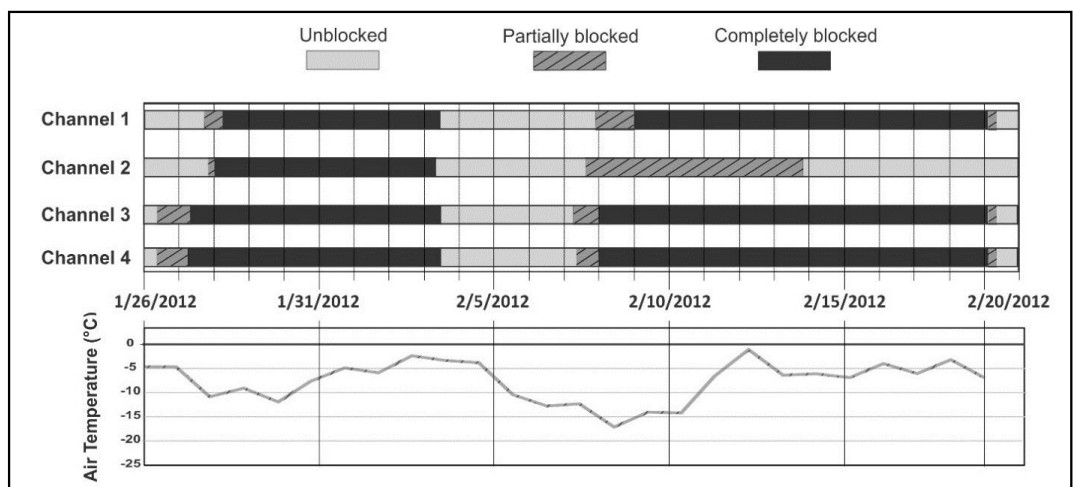

**Figure 3.** Mean daily air temperatures at a gauging station 7 km downstream from the SWIPS and the timings of observed acoustic beam blockages during the Jan. 26-Feb. 20, 2012 time period. Temperature data extend up to and just beyond the upstream advance of the ice edge past the SWIPS site. Uncertainties in blockage transition timings were greatest (+/- 2 hours) at the starts and ends of partial blockages. Brief periods of partial blockage embedded in the completely blocked periods were not included in the plots.

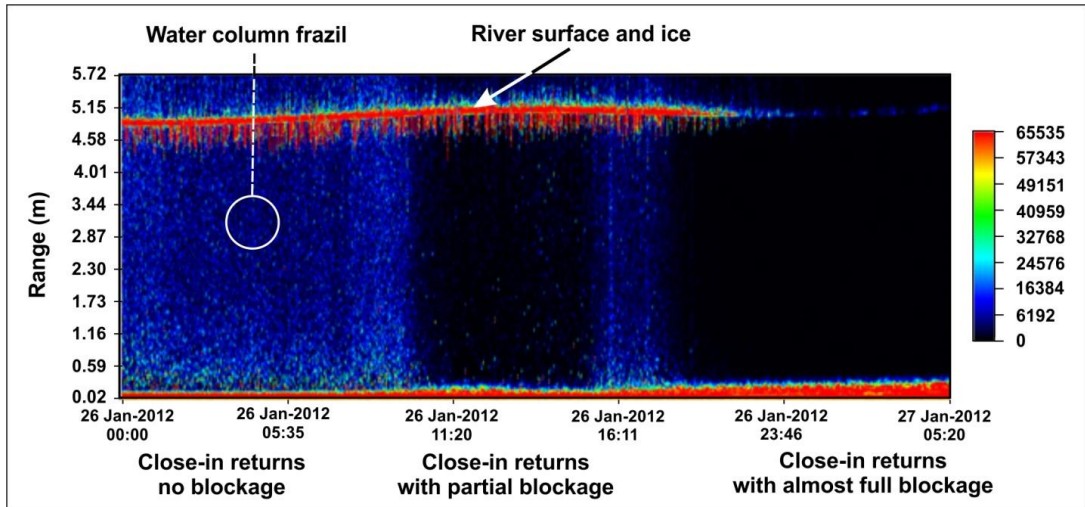

**Figure 4.** Echogram of the channel 4 (774 kHz) returns during the first blockage event.

The single peak $F(t)$ results plotted in Fig. 2 for Intervals 4 and 5 include comparisons with corresponding outputs from anchor ice-free CRISSP1D simulations similar to those reported previously (Jasek et al., 2011). As in the earlier work, these comparisons were necessarily limited by reliance (Section 2.2) on model inputs of water temperature data acquired 370 km upstream of the SWIPS monitoring site. Consequent differences in the timings of, respectively, the simulated and observed onsets of frazil growth ranged between 3.5 to 15 hours for the documented 2011-2012 frazil intervals. Fortunately, it can be shown (Marko et al., 2017) that the close similarity of atmospheric conditions during the actual and the temporally closest significant simulated event still allowed meaningful comparisons. This approach simply shifted the





simulated $F(t)$ output ahead or back in time to bring the simulated and observed frazil onsets into coincidence. The resulting $F_s$(CRISSP) curves in Fig. 2a,b , thus, represented model inputs corresponding, respectively, to times 11 h later and 3.5 hours earlier than the times indicated on the horizontal plotting axes. The second approach, applicable to simulations predicting premature (early) frazil onsets, artificially

incremented the model's upstream water temperature inputs, $T_w$, for a short period sufficient to force simulated frazil nucleation into coincidence with the observed onset. For Interval 5, this adjustment required a brief 0.25°C temperature increase. This artifice insured that all environmental inputs employed in producing the model output, $F_{cont}$(CRISSP) during the observed frazil event, were completely contemporary with the measured quantities, $F$(meas). Comparisons of the shifted, $F_s$(CRISSP) and contemporary,

$F_{cont}$(CRISSP) curves in Fig. 2b confirm expectations that simulated $F(t)$ magnitudes and time dependences were relatively insensitive to finer cooling period details.

**Table 2**. Comparisons of pre-onset heat fluxes and post-onset latent heat fluxes.

| Interval | Pre-onset sensible heat flux ($Wm^{-2}$) from $dT_w/dt$ | Latent heat flux ($Wm^{-2}$)from $dF/dt$ | Latent heat flux/ sensible heat flux |
|---|---|---|---|
| 1 | 162 | 8.9 | 0.07 |
| 3 | 628 | 28.8 | 0.06 |
| 5 | 289 | 29.0 | 0.10 |

It is to be noted that the comparisons in Fig. 2a,b (and in Fig. 5a,b below) required use of $F_s$(CRISSP) and

$F_{cont}$(CRISSP) plotting scales which were 40 times larger than those used to display $F$(meas) results. This difference was necessary to account for the simulations' consistent tendency to over-predict frazil content by as much as 2 orders of magnitude. Specifically, the ratios of simulated to measured quantities ranged, roughly, between 50 and 150 over the course of Interval 4. In Interval 5, the 40-fold difference in the simulated and measured plotting scales was more consistently reflective of observed differences. Most

importantly, the observed characteristic forms of the single peak events, with initial sharp rises being followed by sharp drops and lower, relatively constant, fractional volumes, were not anticipated by the simulations. Instead, the simulated rapid initial rises in fractional volume were followed by persistence of high frazil content, usually for full event duration. All these results supported our premise that model simulations which ignore anchor ice growth are not representative of Peace River frazil behaviour.

Possibilities for addressing this shortcoming were previously explored by Jasek et al. (2011) based upon comparisons between measured surface ice parameters and their simulated counterparts derived from CRISSP1D model runs incorporating anchor ice growth. The lack of consistency in the obtained results combined with the apparent agreement achieved between anchor ice-free models and available surface ice data led to the conclusion that anchor ice was not a major factor in river ice cover growth. In retrospect,

this conclusion was, in large part, a consequence of the model assumption that anchor ice growth occurred primarily by riverbed capture and accretion of water column frazil. Such processes cannot produce significant anchor ice with suspended frazil concentrations at the levels plotted in Fig. 2. On the other hand, the 1 $ms^{-1}$ and larger velocities, typical of most rivers, greatly enhance rates of heat removal from riverbed-fixed ice crystals (Pietrovich, 1956) relative to free-drifting frazil. This difference favours *in situ* riverbed

anchor ice production: allowing interpretation of the results in Table 2 as consequences of neglect of such production as the dominant source of latent heat input to the water column. In the best-documented Interval 5, the Table results suggest that "missing" latent heat contributions from this source were equivalent to a solid ice layer thickening at a rate of 2.9 $mmh^{-1}$. This estimate, and the underlying mismatch of sensible-latent-heat fluxes, tells us much about the dynamics of the critical early stages of seasonal ice growth.

Specifically, it would appear that, apart from being initially "seeded" by captured frazil crystals, riverbed anchor ice growth controls rates of frazil production and not the other way around as assumed in the Jasek et al. (2011) simulations.





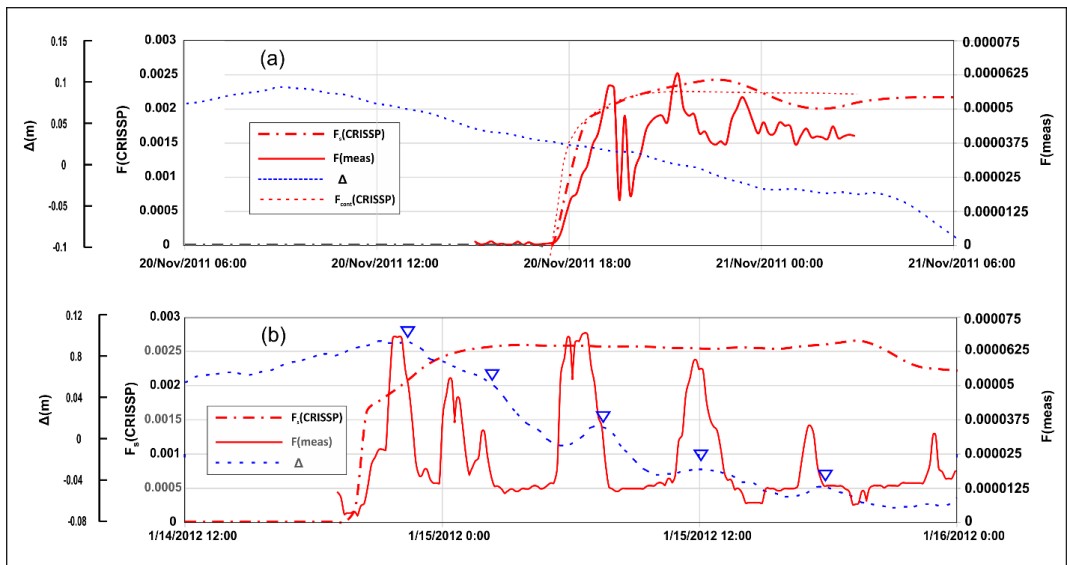

**Figure 5a,b.** Comparisons of fractional volumes as measured ($F$(meas)) and simulated ($F_s$(CRISSP), ($F_{cont}$(CRISSP)) for a) Interval 1 and b) Interval 3. The $F_s$(CRISSP) plots were shifted to later and earlier times by 12.65 and 15 h, respectively, to facilitate coincident simulated and measured frazil onsets. The plot in a) also includes $F_{cont}$(CRISSP) depicting an artificially triggered simulation described in the text based upon fully contemporary post-onset environmental inputs. The measured and simulated fractional volumes represented, respectively, regions 2.3 m above the SWIPS instrument and water column mean values. The additional, dotted, curves represent $\Delta$, the difference between 10-point running- average local water levels and levels simulated by CRISSPID in the absence of anchor ice. The inverted triangles in b) denote the central positions of anomalous anchor ice-related water level peaks.

Efforts to account for finer details of the 2011-2012 acoustic field data provided a basis for expanding this conclusion into a quantitative interpretative model of lower water column ice processes. For example, the time delays separating the first onsets of frazil formation from subsequent initial signs of acoustic blockage were found to be linked to rates of riverbed ice layer growth. A second source of data on such layer changes was encoded in the time dependences of the "close-in" acoustic returns depicted in the Echogram of Fig.4. The range spanned by this feature, associated with backscattering by ice on or immediately above the transceivers, was observed to increase in concert with attenuation and extinction of acoustic returns from the water column and river surface. Such changes contained basic information on growth rates in those portions of the anchor ice layer positioned above the SWIPS transducer.

Studies of these more subtle aspects of the collected data were feasible because of applications of electrical heating to the instrument package. This heating was intended to assure the physical stability of the instruments and their capabilities for water column profiling. The sudden losses of such capabilities several hours into Intervals 4 and 5 were, initially, inexplicable. However, the above-cited thermodynamic evidence for growth of anchor ice volume at rates of several mmh$^{-1}$, raised possibilities for eventual impacts on acoustic measurements made with sensing elements 29 cm above the riverbed. As noted above, such impacts, the initial stages of acoustic blockage, were observed in Intervals 4 and 5, 15.0 and 6.0 hours, respectively, after first onsets of frazil growth. These delays were interpreted as the times required for the upper surface of a growing riverbed layer of porous anchor ice to reach the transducer faces. Consequent accumulation and "overspilling" of this ice layer onto and above the transducers was likely to resemble behaviour previously reported by Qu and Doering (2007) for anchor ice produced by frazil capture in a laboratory flume. Given the elevation of the transducer faces and the inferred 2.9 mmh$^{-1}$ growth rate of an





equivalent layer of solid ice, these results suggest that, during, at least, the first 6 h of Interval 5, an anchor ice layer, with a porosity of, roughly, 90%, was accumulating at a rate of, roughly, 4.4 cmh$^{-1}$.

The second source of additional information allowed quantifying the stages of layer growth in which anchor ice was already in contact with or above the SWIPS transducers. This ice not only weakened or
eliminated acoustic returns from water column frazil and surface ice but also contributed its own "close-in" backscattered returns. The parameter of interest in documenting this ice was the extent to which the range spanned by the close-in feature was incremented above its minimal (baseline) value as observed prior to any weakening of acoustic returns from the main body of the water column. As seen in Fig. 4, such increments in close-in return width typically varied during a frazil event, with initial widening being
reduced or eliminated during warmer mid-day hours before increasing again, accompanied by the fading and disappearance of water column and surface returns. Meaningful estimates of thickness increments were only possible during periods associated with thickening of this feature. (Such thickening confirms that the acoustic pulses are still penetrating and, hence, sampling the full height of the close-in ice layer.) Our procedure estimated such range increments for portions of the close-in returns having digital signal levels
> 24,000 counts as a function of time after the first signs of return blockage. This treatment ignored the noted temporary decreases in close-in range span: presumably caused by melting and/or ice detachment. While probably underestimating actual thicknesses, the use of a moderately high-count threshold minimized possibilities for thickness overestimation introduced by late returns associated with multiple scattering (acoustic paths involving more than one scattering target).

Increments in the thickness of the close-in ice layer, $\Delta r$, were derived from corresponding increments in, $\Delta r'$, the maximum range of associated signal returns. The latter quantities, which could be read off the zoomed-in Echogram scale or extracted from the underlying digital data, required conversion to ice thickness with the simple relationship:

$$\Delta r = \left( \frac{c_a}{c_I} \right) \Delta r' \qquad . \qquad (6)$$

where $c_I$ and $c_a$, respectively, denote the speeds of sound in freshwater and anchor ice at 0°C. Reasonable minimum estimates for $c_a$ equal to 1000 ms$^{-1}$ and 1200 ms$^{-1}$ were available, respectively, from earlier laboratory- and Peace River field-measurements on slush ice (Marko and Jasek, 2010a). A more robust, if less relevant, extreme upper limit was given by the 3840 ms$^{-1}$ value measured by Vogt et al. (2008) in bubble-free zero porosity ice. Using these results to set 1000 ms$^{-1}$ and 1800 ms$^{-1}$ as possible lower and upper
bounds of anchor ice sound speed, it was convenient to set $c_a = c_I$, allowing $\Delta r$ and, hence, close-in thicknesses to be read directly off Echogram plots. This assumption was consistent with the high porosity deduced above from the timings of blockage onsets, giving tolerable +/-30% uncertainties in thickness estimates. To minimize complications from wavelength-sensitive near-field effects and transducer ringing, thickness estimates were derived from data acquired at the highest acoustic frequency, 774 kHz (channel
4), characterized by the lowest pre-blockage ranges spanned by close-in returns. The plotted results (Fig. 6) showed detectable accumulations reaching 23 cm and 14 cm during, respectively, the first and second blocking events. Roughly similar, 3-4 cmh$^{-1}$, growth rates characterized the initial stages of both events which preceded temporary clearances and, thus, offered the best fundamental measures of layer thickening rates. Thus, at least prior to thicknesses becoming large enough rule out further measurements, layer growth
rates at levels $\geq$ 30 cm above the riverbed were comparable to those inferred from earlier stages of frazil and anchor ice growth.





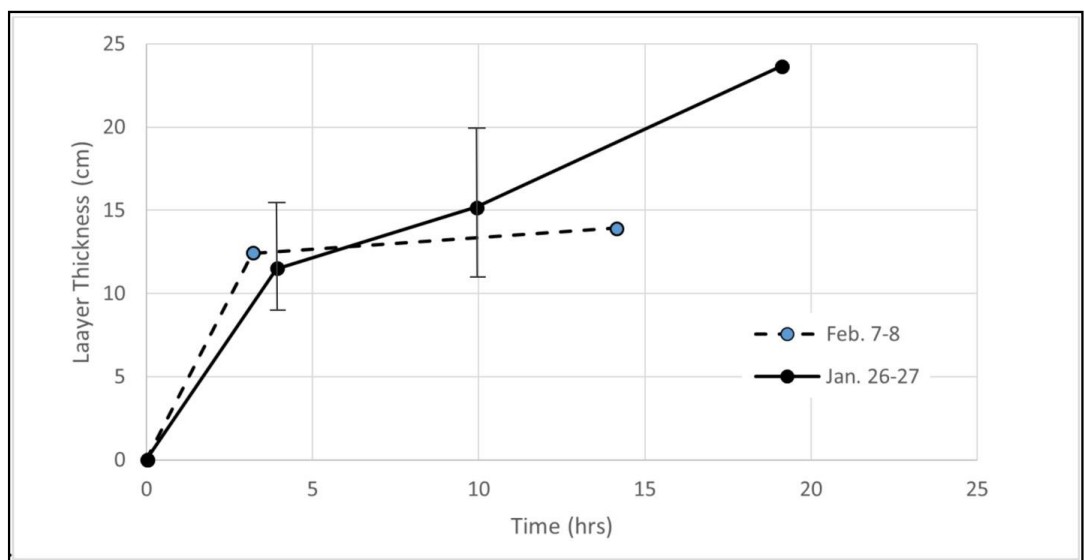

**Figure 6.** Estimates of anchor ice accretion above the elevated SWIPS transducers during the Jan. 26-27 and Feb. 7-8 blockage events as a function of time since event onset. The Jan. 26-27 error bars data are representative of both Intervals.

5      Evidence was also acquired suggesting significant growth occurred at layer thicknesses which exceeded those plotted in Fig. 6 but which could not be quantified due to evidence of incomplete acoustic penetration of still higher portions of the ice layer. Instead, it can be seen (Fig. 7) that the inferred continuation of growth over periods several days in duration produced systematic changes in the temporal and spatial spectra of returns from the insonified lower portion of the ice layer. The observed changes were consistent

10    with continued layer growth and are illustrated in the Figure in successive panels representing data acquired over three 13 h periods separated by 24-hour intervals. The panels show significant reductions with time in the spatial resolution of image features representing variations in return signal strength. This trend toward greater homogeneity was indicative of increasing ice layer stability. In the last third of the last panel, high frequency variations suddenly reappeared, presaging imminent layer clearance.

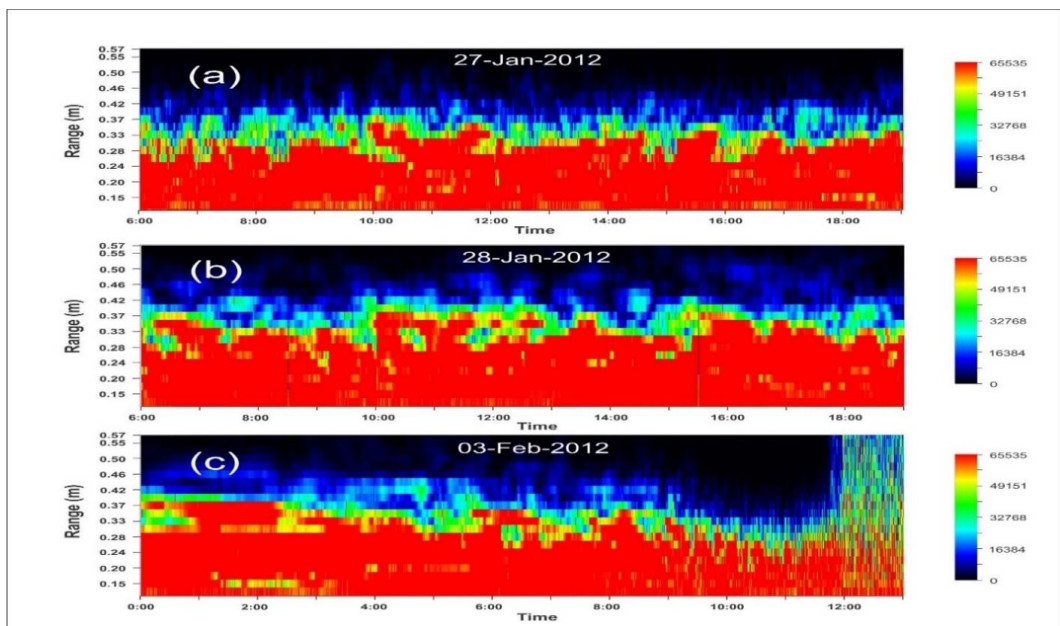

**Figure 7.** Zoomed-in channel 4 close-in returns acquired during 13 h intervals at different stages of the Interval 4 blockage event. Panels a) and b) depict intervals initiated at times coinciding with complete extinctions in, respectively, the high (06:00, Jan. 27) and low (06:00, Jan. 28) frequency channels, roughly 24 and 48 hours into the interval. Panel c), initiated at 00:00, Feb. 3, terminates with a blockage-ending ice clearance event.

Additional single peak analyses were carried out to quantify connections between *in situ* frazil growth and environmental forcing. Results obtained during Interval 5 were of particular value in this respect: effectively offering a basis for estimating the thermodynamic requirements for anchor ice growth sufficient to initiate acoustic blockage. The special appeal of this particular interval was that it was immediately preceded by 72 hours of undetectable water column frazil during a period in which simulated- and ADCP-measured water temperatures rose as high as 1.1°C and 0.5°C, respectively. Under such conditions, subsequent growth of an anchor ice layer could be assumed to have begun on a completely ice-free riverbed at the observed 23:30, Feb. 6 onset of frazil growth. Consequently, the 6 hour delay separating the latter onset from the first detectable signs of acoustic blockage offered a means of estimating a minimum cumulative heat flux requirement for producing the 29 cm layer of 90% porosity ice postulated above to have initiated the observed beam obstruction. CRISSP1D cumulative heat fluxes, $\Phi$, were calculated for the unblocked portion of the frazil Interval using contemporary atmospheric and water temperature inputs, $T_a$ and $T_w$ and a linear proportionality between $\Phi$ and water-air temperature differences. This flux, can be expressed as:

$$\Phi = K\ (T_w - T_a)(1 - C_i) \qquad , \qquad (7)$$

where $C_i$ represents surface ice coverage and $K = 17$ Wm$^{-2}$ denotes a proportionality factor established by Shen et al. (1984) from optimal matching of modelled and measured surface ice parameters. Plots of air temperature data and modelled heat flux results showed (Fig. 8) that the onset of blockage corresponding to the end of the frazil Interval (denoted by the shading) required a cumulative flux of 5.6 MJm$^{-2}$.




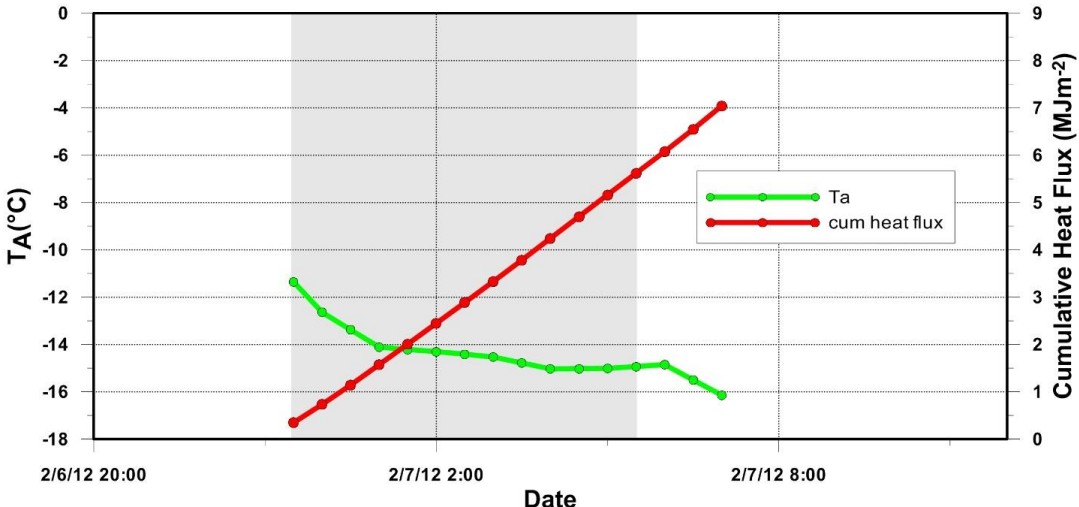

**Figure 8.** Plots of CRISSP1D air temperature inputs and unshifted cumulative heat fluxes for Interval 5. The timing of the Interval is denoted by shading.

Hydrostatic pressure data collected on the ADCP instrument provided additional information on river and
ice conditions. Unfortunately, connections between riverbed ice and water levels are, generally, ambiguous
(Beltaos, 2013), especially in large regulated rivers subject to multiple sources of variability. Water level
variations, driven by anchor ice-induced changes in river cross section and bed roughness can be of either
sign, depending upon the stage and the relative positioning of the ice growth and the measurement site
(Jasek et al., 2015). Kerr et al. (2002) have reported similar sensitivities. Ice impacts on these variations
were documented by comparisons of measured water levels with expectations from ice-free CRISSP1D
simulations which applies corrections for flow travel times and attenuation of regulated dam discharges.
The ice-related water level changes, plotted in Figs. 2a,b, show that onsets of the two single peak Intervals
roughly coincided with the beginnings of slow, sustained level rises. The Interval 4 increase peaked at 0.6
m (Marko et al., 2017) just prior to the Feb. 3 clearance event (Figs. 3 and 7), significantly higher than the
10 to 20 cm increases associated with earlier frazil events.

Taken together, these independent and relatively coarse characterizations of riverbed ice formation suggest
that *in situ*-grown anchor ice is the dominant species of subsurface river ice. In this view, the most important
function of water column frazil is to provide seed crystals which adhere to the riverbed, initiating *in situ*
anchor ice growth and its accompanying latent heat production. The latter heat reduces frazil growth below
levels otherwise anticipated, and, in single-peak events, frazil production is primarily limited to a relatively
short-lived burst of growth as an approximate equilibrium is established, at much lower levels of frazil
content, with latent heat from ongoing anchor ice growth. This equilibrium is maintained until $F(t)$ begins
to fall at the ends of the plotted sequences (Fig. 2), signifying onsets of beam blockage. Relatively brief
reversals of the blockage process, triggered by diurnal atmospheric variations, were readily detectable (Fig.
4). Evidence that such reversals release large anchor ice fragments was available from Echogram data
showing sudden reductions in the widths of close-in ice features and the appearance (Marko et al., 2017) of
very strong, short-lived (one ping) targets in the water column during such periods. Video data (Jasek et
al., 2015) have confirmed sudden surfacing of floes with linear dimensions measured in metres. More
recently, Evans et al. (2017) used sidescan sonar to map spatially extensive riverbed ice layers with
thicknesses on the order of or exceeding 10 cm.



### 3.2.2 Multi-peaked frazil growth intervals

The results presented in Figs.5a,b for the November and mid-January Intervals 1 and 3 offered striking contrasts with the above-described single peak behaviour. Differences included both the oscillatory multi-peaked form of $F(t)$ and a tendency for water levels to peak at or shortly before frazil onset and, subsequently, decrease. Mean air temperatures for these two Intervals, -18°C and -20°C, were significantly below those, -6°C and -15°C, associated with Intervals 4 and 5. Again, the limitations of CRISSP1D input data required comparisons which shifted simulated $F(t)$ outputs 12.6 h back and 15 h forward, respectively, for Intervals 1 and 3 to force coincident simulated and observed frazil onsets. As in Interval 5, a premature simulated onset of Interval 1 allowed use of a brief artificial, 2.1°C, increase in upstream dam water temperatures to effect comparisons of $F$(meas) with fractional volumes, $F_{cont}$(CRISSP), simulated with fully contemporary environmental data.

Again, simulations neglecting anchor ice production failed to capture both the magnitudes and qualitative character of the observed variations in frazil contents. The colder air temperatures accompanying multi-peaked frazil production increased the peak magnitudes of frazil fractional volume by, roughly, 20% relative to the single peak intervals. Nevertheless, the principal impacts of lower temperatures appeared to be the introduction of repetitive oscillations corresponding to appearances of additional strong narrow $F(t)$ peaks usually separated by longer periods of relatively steady lower frazil content. In all cases, frazil content, again, fell short of simulation expectations by more than one order of magnitude.

This behaviour lends itself to interpretation in terms of the *in situ* anchor ice growth model developed above for single peak behaviour and acoustic blockage occurrences. In fact, it is not unreasonable to anticipate that much of the distinction between the two generic types of frazil intervals can be attributed to differences in the external environmental energy inputs which were shown to control riverbed ice growth and its subsequent upward transport to the river surface. Exploring this possibility can draw upon two important pieces of information: namely, 1) that acoustic blockages were not observed during the multi-peaked frazil intervals; and 2) frazil growth sufficient to initiate acoustic blockages was estimated above to require cumulative fluxes of, at least, 5.6 MJm$^{-2}$. The latter requirement explains the absence (Marko et al., 2015) of blockages during an additional 6-hour, single-peaked, frazil interval (Interval 2 in Table 1) during which the accompanying cumulative flux over the full Interval, (calculated using Eq. 9), was only 2.8 MJm$^{-2}$. Similar absences during the multi-peaked Intervals 1 and 3, coincided with air temperature ranges of -16.5°C to -19°C and -14°C to -22°C, respectively. Cumulative heat flux calculations indicated that, at the recorded air temperatures, exceedance of the 5.6 MJm$^{-2}$ threshold for detectable blockage would have occurred 5.2 h (Interval 1) and 4.7 h (Interval 3) after the resumption of anchor ice growth at the termination points of the initial $F(t)$ peaks in these Intervals. However, the observed appearances of additional frazil peaks 3.7 and 4 hours, respectively, after the corresponding initial Interval peaks suggest that the latter peaks had been immediately preceded by new clearances and transports of released anchor ice to the river surface. These releases would have short-circuited the ice layer buildup at thickness values below the threshold required for initiating acoustic blockage. The observed additional peaks were, thus, markers of re-accelerated frazil growth in response to the lower rates of latent heat production characteristic of a recently partially or fully depleted anchor ice layer. In this interpretation, the multi-peak structure of frazil events is a consequence of alternating intervals of layer growth and clearance controlled by the time dependent physical stabilities of rapidly growing riverbed ice layers. The results suggest that, in Intervals 1 and 3, layer growth tended to vary drastically on temporal scales shorter than the 6.0 h and 15.0 h periods required, in Intervals 4 and 5, respectively, to initiate acoustic blockages. The implied differences in the stabilities of the riverbed ice during the alternative single- and multi-peaked- interval varieties are most easily explicable in terms of ice layer stability being very sensitive to the air temperatures which accompany layer growth. This possibility was first suggested by Parkinson (1984) who identified physical differences between anchor ice grown at, alternatively, air temperatures between -5°C and -10°C and near -20°C. More recently, Dube′ et al. (2013) explicitly noted that the physical stabilities of ice dams containing anchor ice tended to increase when grown during multiple intervals of modest cooling as opposed to single episodes

of "hard" freezing conditions.  It is, thus, to be expected that, in the latter case, i.e. layer growth at air temperatures below, very roughly, -15°C, a riverbed ice layer is much less likely to remain in place relative to one grown at higher temperatures ("soft" freezing conditions). This situation is schematically depicted In Fig. 9.

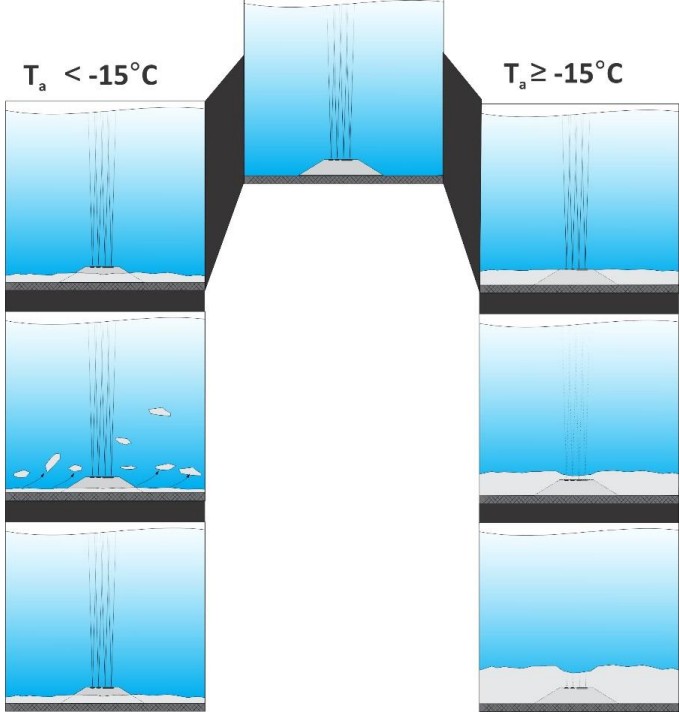

**Figure 9.** Schematic illustrations of impacts on SWIPS profiling of the proposed anchor ice layer evolution mechanism (with time advancing downward) under alternatively soft ($T_a \geq$ - 15°C) and hard ($T_a <$ - 15°C) supercooling conditions.

Nevertheless, aside from differences in the relative stability of the respective anchor ice layers, distinctions
between single peak and multi-peaked frazil events may still be possible, within our limited data base, in terms of the opposing signs of the water level changes during frazil growth. Thus, the steady decreases noted in ice-related water level displacements, $\Delta(t)$, during both multi-peaked Intervals (Figs 5a,b) ran directly counter to the rising trends noted during single peak Intervals (Fig. 2a,b). Interpretations of these differences are complicated by the above-noted sensitivities of local water level responses to: riverbed
conditions; stages of development and measurement locations (Kerr et al., 2002: Jasek et al., 2015). A possible exception to this complexity was observed during Interval 3 in the form of weak but detectable peaks in $\Delta(t)$ which consistently appeared to coincide with the downslopes of $F(t)$ peaks. This timing was compatible with the above interpretations of oscillatory fractional volume behaviour and could be taken as evidence of the re-starting of "new", hydraulically "rough", anchor ice growth (Kerr et al., 2002) after
partial or complete layer clearances. The latent heat produced by this new growth would have already reduced supercooling enough to force reductions from the immediately preceding fractional volume peak value and continuing growth could begin to smooth ice layer topography, eliminating the observed initial small water level elevations. Efforts to test, modify or replace this hypothesis may be hard to justify given: the brevity and magnitude of the effect and its confinement to single frazil interval which was associated
with the seasonally maximum cooling rate.



## 4. Summary and Implications

### 4.1 Summary

The results presented in this paper were based upon data acquired during a single annual field program in one river. Given the central role of numerical river ice models and, specifically, the CRISSP1D model, in the management of that river, a primary goal of the data collection and analysis work was model "calibration". It soon became apparent that the quality and breadth of the acquired data allowed more than fine tuning of model parameters. Specifically, measured peak frazil fractional volumes were, at least, 50 times smaller than inferred from model interpretations of surface ice growth data. Moreover, while high fractional volumes were anticipated throughout the durations of frazil events, frazil concentration estimates were usually about 50% to 80% below levels attained in brief periods of maximal content. Data on the timings and volumes of instrument ice accretion and consequent losses of profiling capabilities were accidental, but essential, inputs for understanding deviations from expectations.

These understandings utilized a simple interpretative model which assumed that supercooling triggers near-simultaneous production of frazil in the water column and *in situ* growth of anchor ice on the riverbed. The latter growth is, almost certainly, initiated by relatively small numbers of frazil crystals which previously impact upon and adhere to the riverbed. This situation reflected the large disparities in the rates of turbulent heat dissipation at, alternatively, the surfaces of free-drifting- and static, bottom-affixed-frazil crystals. The larger relative water flow velocities in the latter case made the riverbed the principal site of sub-surface ice production. Thermodynamic data and SWIPS measurements of suspended frazil fractional volumes suggested that, in the initial, most intense, portion of a frazil event, riverbed ice mass growth exceeds water column frazil mass production by more than an order of magnitude. Given this disparity and the resulting dominant influence of latent heat production by anchor ice on water temperatures, it is not surprising that river models neglecting *in situ* growth greatly overestimate frazil concentration. Frazil variations closely track changes in the riverbed ice layer.

Although it was convenient to give separate considerations to frazil intervals characterized by two apparently different generic forms of variability (single-peaked and multi-peaked), it was eventually concluded that these distinctions, primarily, reflected differences in the physical stabilities of the corresponding contemporary anchor ice layers. In spite of the small number of analyzed intervals, our results and prior observations by other workers (Parkinson, 1984; Dube′ et al., 2013) suggest that layer stability decreases proportionately with increasing cooling rates. Specifically, air temperatures below -15°C appeared to favour growth of unstable ice layers which give rise to repeated peakings in suspended frazil content following buoyancy-driven movements of high porosity anchor ice to the river surface.

Estimates suggest that riverbed production of anchor ice mass per unit area of riverbed was equivalent to accretion rates of approximately 3 mmh$^{-1}$. This accretion occurs in the form of high (90%) porosity layers growing at a rate of, approximately, 3-4 cmh$^{-1}$. The detected appearances of anchor ice in thicknesses of, at least, 25 cm or more on or just above instrument surfaces almost 30 cm above the riverbed suggests that extended periods of moderate supercooling, can produce riverbed anchor ice layers with overall thicknesses approaching or exceeding 0.5 m. Changes in these layers have direct but often ambiguous impacts on water levels. Even spectacular water level change features such as the brief, immediately post-frazil peak elevations highlighted in Fig. 5b were confined to just one out of seven frazil events reviewed in the 2011-2012 data sets (Marko et al., 2015). Nevertheless, it is clear that even a single season's worth of environmental and SWIPS-derived frazil fractional volume data offers a basis for quantitative interpretations of the roles of frazil and anchor ice in seasonal ice cover development. Contrary to prior expectations, the results indicate that *in situ* anchor ice growth and its sensitivities to cooling rates control variations in a river's suspended frazil content.



## 4.2 Implications for past and future research work

Only recently have data analysis and modelling efforts (Jasek et al., 2015; Kempema and Ettema, 2015; Makkonen and Tikanmati, 2018) begun to recognize the importance of *in situ* anchor ice growth processes. It less clear, however, that the dominant role of these processes has been fully appreciated. One consequence
of this situation has been the reduced credibility of modelling assumptions based upon measurements made in laboratory tanks or flumes where *in situ*-grown anchor ice is rarely present. Modelling frazil growth at concentrations similar to those obtained under such unrealistic conditions poses serious non-linearity and self-consistency problems in the presence of, even, comparable volumes of *in situ*-grown anchor ice. Our results suggest that the large volumes of such ice and its accompanying latent heat production may
considerably simplify modelling problems. Indirect evidence for this optimistic view can be found in the highly constrained range of observed frazil contents. Specifically, $F(t)$, typically, varied between 0.001% and 0.01% and appeared to be most commonly associated with a "baseline" value which varied from Interval to Interval between 0.001% and 0.004%. Within our simple model, it is reasonable to suspect that this limited range of variability reflects similar constraints on *in situ* anchor ice growth rates and the
resulting latent heat production. Ultimately, such growth rates are determined, primarily, by atmospheric heat exchanges moderated by possible additional sensitivities to flow velocity and the nature of the riverbed. The measurability and relative stability of these rates suggests that the principal challenge in frazil/anchor ice modelling may be to provide quantitative descriptions of the periods of relatively constant frazil water column content which follow or immediately precede peak frazil presences. Progress in understanding these
periods would allow anticipation of ice layer clearances and representations of initial stages of ice growth on partially or completely ice-free riverbeds. An initial effort to address such clearances was made by Jasek et al. (2015) in terms of "anchor ice waves" describing up-river advances of the downstream boundaries of riverbed ice fields.

The reduced importance of frazil under a developing ice cover, implied by our results, is countered by its
essential role in initiating *in situ* anchor ice growth and, less obviously, by its usefulness as a tool for indirectly monitoring the less visible but more influential anchor ice constituent. The rough estimates of anchor ice growth rates and properties presented above need further verification and refinement to support quantitative treatment in detailed ice models. Such models can be highly local in applications, addressing specific intake blockage or flooding problems. Operationally useful assessment/prediction procedures
would benefit from SWIPS frazil data collection carried out in conjunction with underwater imaging or more sophisticated acoustic techniques capable of quantifying anchor ice accretion patterns.

Recent efforts in the first of these directions by Ghobrial and Loewen (2020) (henceforth referenced as GL) used successive digital images to estimate anchor ice growth rates on an artificial substrate in very shallow (primarily, 0.6 to 0.7 m) river waters. Critical data on water column frazil contents and the status of much
larger adjacent volumes of riverbed anchor ice were not collected although a major objective of this work was to clarify the relative importance of alternative frazil capture- and *in situ*-anchor ice growth mechanisms. These two missing bodies of information were directly relevant to, respectively, assessing the role of frazil ice in the growth process and in accounting for significant differences among the studied events. In the latter case, it was notable that only one of the six observed events coincided with a "classic"
supercooling water temperature curve, usually characteristic of the frazil growth events which accompany anchor ice growth. Nevertheless, the time dependences of anchor ice thickness estimates derived during this and two other events showed a common form in which initial steady, 1.5 to 2.0 cmh⁻¹, growth rates slowed down by, roughly, a factor of two when layer thicknesses reached and exceeded values in the 4 to 8 cm range. These results were not inconsistent with the anchor ice growth depicted in Fig. 6 of the present
paper as derived from acoustic measurements in significantly deeper (5 m) waters. Nevertheless, the change



in growth rates with time (attributed in our case to partial clearances and resumptions of *in situ* ice growth) was interpreted by GL as evidence of transitions between alternative *in situ-* and frazil capture-driven forms of anchor ice growth. This interpretation appeared to be based upon qualitative changes in the appearance of the ice, as imaged, at 5 minute intervals, during these events. Early portions of the growth in the "classic"
event coincided with the initial steep increase and following, similarly steep, decrease in supercooling levels characteristic of such events. The images in this period, which includes, in our model, the peak in water column frazil content, were dominated by large crystals which facilitated unambiguous estimates of growth rates. Subsequent growth after this period occurred under relatively stable and lower "equilibrium" levels of supercooling: yielding progressively more amorphous-appearing ice. This change required growth rates
to be estimated, with lesser accuracy, from the averaged vertical positions of points on the discernable upper boundary of the ice layer. The 0.4-0.9 cmh$^{-1}$ rates deduced in this regime were only slightly larger than those estimated in three other anchor ice events which lacked both evidence of elevated initial growth rates and the crystal structure signatures of *in situ* growth. It was concluded that anchor ice growth in the latter events and in the post-transition portions of the other three study events was a consequence of physical
capture of water column frazil particles. More specifically, this growth was considered to be representative of  accumulations of large numbers of frazil particles adhering to the substrate and each other as opposed to *in situ* growth originating from small numbers of adhering frazil seed particles. The latter process was central to the *in situ* growth-based model developed in our work to address the large imbalances in calculated river/atmospheric energy exchanges which arise when latent heat production is derived from
measured frazil concentrations and laboratory estimates of frazil capture probabilities.

The GL interpretation of the three events exhibiting rate transitions was that the early dominance of *in situ* growth was eventually superseded by the increasing effectiveness of growth arising from frazil particles impacting upon and adhering to prior ice accumulations. Representing time rates of change in layer thickness as a sum of separate frazil capture- and *in situ*-growth terms, the frazil capture term, was assumed
to be proportional to the product of the fractional volume of the suspended frazil population and a capture coefficient, $\gamma$, previously estimated in several laboratory studies. Within this picture, the postulated transitions to frazil capture dominance required significant increases in the values of one or both of the latter parameters. In the absence of specific frazil measurements, the only source of water column frazil content data was, a video display of successive images which showed no evidence of increases with time
in visible concentrations of freely floating frazil particles. Evaluations of the alternative explanatory option, namely, occurrence of sufficiently large values of the frazil capture coefficient, $\gamma$, were limited to assessing the compatibility of lower bound estimates of this parameter with laboratory-derived values. These calculations utilized an average value of GL- estimated late-event growth rates in conjunction with a frazil fractional volume equal to 1%, which corresponded to the extreme high end of published, laboratory-
derived, values.  It was concluded only that $\gamma$ was "likely not significantly less than $10^{-4}$ ms$^{-1}$". This result, ignored evidence (Marko et al., (2015), confirmed in the present paper, that typical frazil fractional volumes are on the order of 0.002% over the bulk of the durations of frazil events. Making the appropriate change in this parameter raised the estimated lower limit of $\gamma$ to 5 x $10^{-2}$ ms$^{-1}$: a value which far exceeded the upper end of laboratory-estimated parameter values. This obvious discrepancy had previously forced recognition
(Marko et al., 2015) of the reality that river simulation models could not be reconciled with frazil content data without specific inclusions of *in situ* anchor ice growth mechanisms.

Our rejection of the proposed capture-driven mechanism for anchor ice growth highlights the hazards of excluding access to non-laboratory data on one or more interacting components of a complex physical system. Nevertheless, we believe the GL results were useful in confirming the magnitudes of typical anchor
ice variations and in providing evidence for long periods characterized by reduced *in situ* anchor ice layer thickening. Such periods were compatible with our proposed model since such periods appeared to be



contemporary with portions of the frazil growth cycle associated with minimum "equilibrium" levels of supercooling. Greater clarity on this and other possibilities is likely to require achieving greater consistency in the correlations between water temperatures and changes in the frazil and anchor ice river constituents. Much of the observed event to event variations in these respects as well as differences in the forms of rate

time dependences may have been explicable in terms of missing data on the recent history of local water temperatures and ice conditions on the instrument frame and surrounding riverbed. Such information could enlighten choices of measurement timing and interpretation. It is also worth noting that, in the absence of porosity and other information, image-derived results do not easily translate into the ice mass changes which are of most direct relevance for model development. Such data are currently most accessible through

applications of the basic Osterkamp (1978) thermodynamic approach to measured values of water column frazil content and air- and water-temperature. Thus, while more elaborate studies of *in situ* anchor ice growth are to be encouraged, effective research programs still must, inevitably, address all relevant aspects of the physical environment including, in particular, frazil ice in the water column and the underlying thermodynamic regime which links two critical ice species.

**Acknowledgement**

The authors would like to thank David Billenness of ASL for embellishments of RUNSWIPS software and Martin Jasek and BC Hydro for providing access to Peace River field data and CRISSP1D model outputs.

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
