# Peer review of "Analyses of Peace River SWIPS data and its implications for the roles played by frazil ice and *in situ* anchor ice growth in a freezing river"

_The Cryosphere, 2020_

## Referee Comment (RC1) · Anonymous Referee #1 · 25 Nov 2020

**General comments**

This manuscript examined suspended frazil ice and riverbed anchor ice growth in the shallow river based on the acoustic observation. The notable point of this manuscript is that the results and discussions were mainly based on field measurements. Thus, this manuscript is important to understand river ice system. The author compared the measured and modeled frazil ice volume, and showed that the former was quite smaller than the latter. This gap was explained by the latent heat of in-situ reverbed anchor ice growth, which is ignored by the river ice model. The author also proposed

the riverbed and underwater situations for single- and multi-peak frazil growth intervals. These are reasonable qualitatively. Additional quantitative discussions are expected to well understand the winter river ice condition.

I have major comments listed below.

1) For single peak frazil growth intervals, the large difference in measured and modeled value of frazil ice volume was shown. As the author suggested, in-situ riverbed anchor ice growth can be a factor of the difference because the river model ignored it. Did the volume of anchor ice growth on the riverbed reach to the level to explain the gap between measured and modeled frazil ice volume with 1 order of the magnitude quantitatively? Does the model overestimate suspended frazil ice volume in the case of lack of in-situ riverbed anchor ice growth? I would like to see more discussion.

2) The author showed that the river ice model overestimate suspended frazil ice volume. The results and discussions were based on the field data during single peak frazil growth intervals. On the other hand, these cases are not suitable to calibrate the model because of the presence of riverbed anchor ice. Are there some frazil ice growth events without the presence of riverbed anchor ice? If the model is able to estimate suspended frazil ice volume in such cases, anchor ice growth becomes to be a great factor for the model simulation.

3) The author presumed riverbed and underwater situations for single- and multi-peak frazil growth intervals. These situations are consistent with measured variations of frazil ice volume during these intervals. The author suggested that the air temperature is the key factor to induce those two situations. The multi-peak frazil events were induced during the periods of cooler air temperatures. According to the discussion of section 3.2.2, accumulated anchor ice layer became thicker during higher temperature periods. However, the heat loss from the river to the atmosphere becomes larger at lower air temperatures under same wind conditions, enhancing frazil ice and anchor ice growth. I would like to see more discussion on this point.

**Specific comments**

P. 1, L. 15 – 17: "A simple physical model .... river ice volume and mass." I agree with your opinion. In addition to it, I would like to see quantitative discussion in the main text.

P. 4, L. 5 – 6: "Detailed analysis were confined to four of five major supercooling events," Why did the author focus on supercooling instead of suspended frazil ice detected from echogram plots? Cooler river water is lighter than warmer water at the temperature below 4 °C. Hence, frazil ice possibly appeared in the water column when supercooling was not detected on the riverbed.

P. 5, L. 30 – 32: "These runs utilized .... a hydrostatic site approximately 370 km upstream of the SWIPS instrument." Was the hydrostatic site located ∼370 km upstream of the SWIPS instruments site? It seems to be too far to apply the data to input the model calculation. Does the author have some comments about it?

P. 6, L. 3: "Five separate intervals of supercooling" How large was the level of supercooling? I recommend that the author add the level of supercooling at each interval in Table 1.

P. 6. L. 34 – 35: "The timings and intensities of the blockages, .... , are summarized in Fig. 3" I recommend that the author should show the situation of acoustic blockages and the air temperature at the same Figures of time series of F(t) in Figs. 2 and 5 or echogram plots in Fig. 4. Direct comparison of the timings of acoustic blockages with time evolution of F(t) or echogram plots helps us understand what the author described.

P. 7. L. 20 – 33: 1) The author pointed out critical timings such as 08:00 Jan 26, but these are difficult to be found in Fig. 4 accurately. It might be better to show such timings in Fig. 4 using some objects like as triangles.

2) The author mentioned the time evolution of "close-in" returns at the lowest end of the range scale, but it is too small to understand its vertical variation. In particular,
the author explained that suspended frazil ice disappeared from the echogram plots due to the acoustic blockage by anchor ice at ~10:00 - ~16:00. However, the vertical evolution of the layer close to the transducer was unclear in Fig. 4 at that timing. Additional panels to enlarge the range near the transducer and to show the timings of the acoustic blockages (as shown in Fig. 3) help us understand the situations of frazil and anchor ice growth.

3) In section 3.2.2, the author suggested that anchor ice which detached from the riverbed and moved to the river surface was detected with the acoustic instruments. Why was such detached anchor ice not detected in the case shown in Fig. 4? Did the accumulated anchor ice melt and lose the thickness?

4) In Line 27 – 28, the author described "Smaller concurrent reduction were apparent in the strengths of the longest range components o the saturated surface returns." But, I was not able to find this situation.

P. 7, L. 36 – 38: "This pattern . . .. to completely block detection of acoustic returns from water column and surface targets." Does anchor ice covering the transducer prevent return pulse only? Emitted pulse might be prevented by anchor ice?? This is just a comment.

P. 8, L. 19 – 20: "Pre-transition sensible heat fluxes, . . .. change in water temperatures measured on the ADCP instrument." Was the heat loss from the river surface to the atmosphere calculated using atmospheric conditions such as the air temperature, humidity and wind speed? When the water temperature is at the freezing point, the change in the water temperature due to the heat loss becomes to be small. In addition, the water temperature can be changed by advection.

P. 8, L. 25 – 27: What was the heat to transform from the heat loss to the atmosphere if it was not used to form ice?

P. 10, L. 25 – 42: The discussion in this paragraph is interesting. In Line 33 – 38,

particularly, the author proposed good discussion of enhancing anchor ice growth under hydrographic conditions in the Peace River. I would like to see more quantitative discussion, if it is possible. Does the total volume of suspended frazil ice and anchor ice become to be consistent with modeled value of F(t)? According to section 2, the instruments were heated. Is this a factor to suppress anchor ice growth or accumulation on the riverbed?

P. 11, L. 10 – P. 12, L. 2: How large the spatial (horizontal) scale of anchor ice on the riverbed? Was anchor ice distributed around the riverbed with uniform thickness? Is it possible that the instruments promotes/suppresses anchor ice formation and accumulation? Is the discussion described in these paragraphs able to be applied only for the case when instruments are deployed on a riverbed?

P. 14, Eq (7): Why does the heat flux depend on the air temperature only?

P. 14, L. 6 – 23: The author described the impact of river currents on the heat loss of anchor ice in P. 10, L. 28 – 38. Can the author consider this effect to evaluate the cumulative heat flux? The cumulative heat flux of 5.6 MJ/m2 calculated from Eq. (7) may not be suitable to be used as the critical value.

P. 16, L. 4 – 5: "a tendency for water level …." This behavior was only found during Interval 3 in Fig. 5b. Did the author mention about Interval 3 only?

P. 16, L. 5 – 6: "Mean air temperature . . . associated with Intervals 4 and 5." The author described the air temperature for each Interval for the first time here. I recommend to add the panels of time series of the air temperature in Figs. 2 and 5.

P. 16, L. 19 – P. 17, L. 4: 1) The author suggested than the air temperature was a key factor of distinctions between single and multi-peak frazil events. Are there other possible factors such as wind speed and current speed? I think that turbulence is needed to bring lighter and cooler water down to the riverbed. If it is right, much water with lower temperature is brought from the river surface to riverbed. In the fact, single and

multi-peak frazil events occurred during higher and lower temperatures, respectively. However, the relationship between such two situation and the air temperature was not explained.

2) This manuscript indicated that multi-peaked F(t) was attribute to detached anchor ice. I propose that anchor ice can detach at least one time for "hard" freezing conditions at Ta $\geq$ -15 °C. Then, the instruments can detect the resuspended anchor ice during the end of single peak event. Did the instrument show such an event in echogram plots or F(t)?

3) If anchor ice was formed around the riverbed and detached, the instruments detect resuspended anchor ice at several times. This scenario can explain the multi-peak F(t) when ice advection was taken into account. How do you think about it?

4) According to Fig. 9, the height of the instrument is a factor to separate between single- and multi-peak frazil events. Does the author have some idea to express the relationship between the instrument height and the air temperature to distinguish the two situations?

---

## Referee Comment (RC2) · Anonymous Referee #2 · 1 Dec 2020

Review of TC-2020-212: ÂńAnalyses of Peace River SWIPS data and its implications for the roles played by frazil ice and in situ anchor ice growth in a freezing riverÂż

The paper deals with the application of a SWIPS in the peace river and the interpretation of the results. The paper also compares the findings from the SWIPS with modelled data using the CRISSP1D river ice model. I think this is interesting findings, and the application of SWIPS could provide new insight into the formation and transport of frazil and anchor ice in rivers. So, I think this could be a valuable paper for ice researchers. I do think some clarifications is needed in the paper and it could also benefit from a

simplification of the structure and the objectives of the work.

I find some of the text quite dense and detailed and sometimes hard to follow. Four events are singled out for the SWIPS analysis, it is single peak and multipeak events, there is the data from the CRISSP1D model and there are other observations from literature mixed into the discussion in chapter 3 and 4 and also in 1 and 2:

- I miss a clear section of the objectives of the study as a final part of the introduction. What is the main objective? Testing of SWIPS? Determining the relation between in situ anchor ice growth and frazil? Testing the CRISSP1D simulations against SWIPS data? Please guide the reader.

- There is a form of reading guide at the end now which could be improved. This promises something on CRISSP1D in section 2, which only amounts to some info on the setup. A improved version of this would be helpful.

There is a focus in the abstract (line 13-15) and in the introduction (30-33) which have "anomalously" low frazil content compared to the CRISSP1D model which I understand from the text does not simulate the formation of anchor ice formation. If this is the case, I am not sure I understand this comparison and the focus on the differences. If the model does not handle anchor ice properly, I do not see why this comparison is an issue at all unless you want to convey to the model developers that they need to improve their model? Or is there a previously understanding from observations that the formation of anchor ice is not a large part of the ice formation process in the Peace river?

Do the discrepancies between modelled frazil and observed frazil + anchor ice development match in some way?

It could be I am misunderstanding this but in the summary it seems that in-situ growth is a surprising discovery. I thought this was a well-established principle of anchor ice development, particularly in smaller rivers and streams where large quantities of

anchor ice is seen developing while the amount of suspended frazil could be quite low. There are a number of works outlining this mechanism, e.g. Turcotte et al. in several papers.

On page 20 you seem to reject the principle of growth of anchor ice by capture of frazil. This might be the case on a large and deep river like Peace, but I do not think this is the case if you look at anchor ice formation in general. In shallow turbulent streams accumulation (capture) of frazil should be considered, see e.g. Stickler and Alfredsen (2009, Hydrological Processes). But it could be difficult to distinguish these processes at times, and I agree with the need to address this as outlined at the end of section 4.2.

When anchor ice is released from the bottom, is drifting anchor ice captured by the SWIPS? Can this be distinguished from frazil particles? It is indicated in the text, but do we see it on the echograms?

What level of super cooling was observed at each event in table 1? Was this measured locally, if so how? Clarify how these periods were identified (start of section 3.1).

You have water temperature and discharge measured at a site 370 km upstream of the SWIPS. How representative is these regarding the location of the SWIPS, e.g., how well did the model simulate the changes in water temperature over this considerable reach?

Table 2, please clarify the methods used to compute the heat fluxes.

Is figure 8 necessary? Could this just have been left for the textual description?

Page 10: Last paragraph is interesting – could be expanded with quantification.

Page 16: Is the flow the same in the single and multipeak events?

Some minor things:

- Figure 9. Provide a time axis, I think that would enhance the readability of this figure.
- Provide a proper reference to Topham and Marko (2020). It is a discussion paper in

the Cryosphere and could be referenced as such. - Provide a complete reference for Ghobrial et al. 2020 The Cryosphere.

---

## Author Comment (AC1) · 30 Dec 2020

Response to Reviewer 1

Point by Point response

Major comments

1) *For single peak frazil growth intervals, the large difference in measured and modeled value of frazil ice volume was shown. As the author suggested, in-situ riverbed anchor ice growth can be a factor of the difference because the river model ignored it. Did the volume of anchor ice growth on the riverbed reach to the level to explain the gap between measured and modeled frazil ice volume with 1 order of the magnitude quantitatively? Does the model overestimate suspended frazil ice volume in the case of lack of in-situ riverbed anchor ice growth? I would like to see more discussion.*

In the absence of measurements of porosity, neither of our two methods for estimating increases in anchor ice thickness directly quantifies anchor ice volume production rates. Such estimates were only derived from the differences between the sensible heat loss rate immediately prior to frazil onset (derived from the cooling rate) and the latent heat produced by measured increases in frazil fractional volumes. Given that the cooling rates were generally compatible with the measured air temperatures which were also a key input for model simulations of suspended frazil fractional volume growth, our anchor ice volume production rate essentially guarantees that it accounts for the differences between modelled and measured frazil contents. If anchor ice growth were not present and buoyant frazil capture were to be the primary source of surface ice production, an effective model would have had to simulate frazil concentrations 1 to 2 orders of magnitude above measured values. It was precisely this discrepancy that necessitated our inference of dominant *in situ* anchor ice growth.

2) *The author showed that the river ice model overestimate suspended frazil ice volume. The results and discussions were based on the field data during single peak frazil growth intervals. On the other hand, these cases are not suitable to calibrate the model because of the presence of riverbed anchor ice. Are there some frazil ice growth events without the presence of riverbed anchor ice? If the model is able to estimate suspended frazil ice volume in such cases, anchor ice growth becomes to be a great factor for the model simulation*

As far as we can tell, the rate of frazil growth is only a small fraction of that expected from the heat losses inferred from pre-event cooling rate throughout the onset of frazil growth: i.e. we immediately see from the table that initial frazil production is much less than expected from the energy balance requirement. This tells us that riverd and water column ice growth is initiated simultaneously, with the anchor ice growth lagging only long enough to allow for the attachment of a small number of seed frazil crystals.

3) *The author presumed riverbed and underwater situations for single- and multi-peak frazil growth intervals. These situations are consistent with measured variations of frazil ice volume during these intervals. The author suggested that the air temperature is the key factor to induce those two situations. The multi-peak frazil events were induced during the periods of cooler air temperatures. According to the discussion of section 3.2.2, accumulated anchor ice layer became thicker during higher temperature periods. However, the heat loss from the river to the atmosphere becomes larger at lower air temperatures under same wind conditions, enhancing frazil ice and anchor ice growth. I would like to see more discussion on this point.*

The reviewer's point is well taken and we believe the consistent association of multiple peak events with lower air temps and greater cooling supports our argument for this view. Nevertheless, it is based upon

observations of a total of two events. However, we also were able to include references to crystal size and the strength of ice dams under soft rather than hard cooling conditions. This interpretation is also compatible with our experience in the growth of high quality metal and semiconductor crystals in which rapid growth increases defect density and structural weakness.

**Specific comments**

*P. 1, L. 15 – 17: "A simple physical model . . .. river ice volume and mass." I agree with your opinion. In addition to it, I would like to see quantitative discussion in the main text.*

The model is summarized for single peaks at the end of 3.2.1. Important features deduced from multipeak data are added in 3.2.2. It is extended in 3.2.2 and an integrated summary is provided in Section 4.1 (p18).

*P. 4, L. 5 – 6: "Detailed analysis were confined to four of five major supercooling events," Why did the author focus on supercooling instead of suspended frazil ice detected from echogram plots? Cooler river water is lighter than warmer water at the temperature below 4 ◦C. Hence, frazil ice possibly appeared in the water column when supercooling was not detected on the riverbed.*

We make no distinction in the text between "frazil" and "supercooling" events: viewing them as interchangeable since frazil formation required supercooling. When frazil appears, it is shows up in comparable amounts throughout the water column.

*P. 5, L. 30 – 32: "These runs utilized . . .. a hydrostatic site approximately 370 km upstream of the SWIPS instrument." Was the hydrostatic site located ~370 km upstream of the SWIPS instruments site? It seems to be too far to apply the data to input the model calculation. Does the author have some comments about it?*

Yes, 370 km was very far from the SWIPS site and, yes, that distance complicated model/measurement comparisons. In fact, as we make clear in our discussions of the individual events, it usually necessitated time shifts and other adjustments in the assumed environmental forcing to allow correlations with the timing of the observed frazil events. The measurements were part of a BC Hydro's annual river monitoring program which was just one of similar programs carried out both prior to and after the 2011-2012 work. The levels of effort were judged to be sufficient to support development and calibration of ice models for hydroelectric and flood management purposes. Clearly the detail and precision of the model comparisons could have been improved with additional, closer, upriver measurements but we do not believe this would have qualitatively altered the obtained levels of model/measurement agreement or our key conclusions regarding the roles of frazil and anchor ice. We elaborate briefly on the underlying robustness of such comparisons in the revised manuscript.

*P. 6, L. 3: "Five separate intervals of supercooling" How large was the level of supercooling? I recommend that the author add the level of supercooling at each interval in Table 1.*

As noted in the text, heating of the instrument package was necessary to prevent immediate blockage of the acoustic beams  and package destabilization. This precluded local measurements at accuracies sufficient for tracking levels of supercooling. The water temperature data were primarily used to detect initial supercooling and estimate the rate of cooling .

*P. 6. L. 34 – 35: "The timings and intensities of the blockages, . . .. , are summarized in Fig. 3" I recommend that the author should show the situation of acoustic blockages and the air temperature at the same Figures of time series*

*of F(t) in Figs. 2 and 5 or echogram plots in Fig. 4. Direct comparison of the timings of acoustic blockages with time evolution of F(t) or echogram plots helps us understand what the author described.*

The only significant blockages that occur during the portions of the F(t) records plotted in Figs 2 and 5 occurred at the ends of the records in Fig. 2 and are noted in the captions. These blockages imtroduce the step falloff in F(t) at the end of the displayed record. It is important to note that F(t) in periods of partial blockage are of little use for assessing water column frazil content. Meaningful content measurements require absence of anchor ice above the transducers.

*P. 7. L. 20 – 33: 1) The author pointed out critical timings such as 08:00 Jan 26, but these are difficult to be found in Fig. 4 accurately. It might be better to show such timings in Fig. 4 using some objects like as triangles.*

We have added a 08:00 marker arrow just below the time axis. There is at least a '/+/- 0.5 hour uncertainty in establishing the first faint trace of the blockage onset. Actually, judgement on this timing also should take into account information provided by the the terminating drop in F(t) in those intervals which incur blockage.

*The author mentioned the time evolution of "close-in" returns at the lowest end of the range scale, but it is too small to understand its vertical variation. In particular, the author explained that suspended frazil ice disappeared from the echogram plots due to the acoustic blockage by anchor ice at ~10:00 - ~16:00. However, the vertical evolution of the layer close to the transducer was unclear in Fig. 4 at that timing. Additional panels to enlarge the range near the transducer and to show the timings of the acoustic blockages (as shown in Fig. 3) help us understand the situations of frazil and anchor ice growth.*

We don't think this is practical and would complicate an already complex Figure. The changes on the display scale are very slight but discernable by the trends even if the precise points where close-in thickening begins are hard to establish from the figure and, as indicated above, should also reflect the F(t) curve.. We do introduce an expansion of the close-in region in a later figure to illustrate important changes which occur in this layer when physically significant accumulations of blockage ice are present.

*In section 3.2.2, the author suggested that anchor ice which detached from the riverbed and moved to the river surface was detected with the acoustic instruments. Why was such detached anchor ice not detected in the case shown in Fig. 4? Did the accumulated anchor ice melt and lose the thickness?*

Our brief explanations may have led the reviewer to misunderstand the limitations of our measurement technique. We are measuring, at any given time after emission of an acoustic pulse, the sum of all returns from a roughly 4 cm horizontal slice of the water column. (The thickness of this slice is essentially determined by the duration in time of the emitted pulse) The returns used in our F(t) extractions are associated with a range cell 2.3m above the transceiver and thus, arise from circular disk-shaped volumes about 0.5m in diameter representative of returns from the mid-water regions of interest for our analyses. The ProfileView plots give measures of the intensities of the returns of a single acoustic pulse from imaginary adjacent disks stacked at ranges between the transducer faces (which give the close- in signals) all the way up to the river surface which gives the river surface returns. When a piece of anchor ice becomes detached and moves upward to the surface it moves through this imaginary pile of stacked disks as the targets move down river at typical 1.25 m/s speeds. For fragments with horizontal dimensions on the order of 1m, the strong returns would be expected to be confined to just one 4cm thick disk…in one pulse… Subsequently the ice would be completely out of the beam and, probably, in a different range when, 1 s later, the SWIPS instrument emits and detects another pulse. The only way of acoustically seeing such fragments would be in the Profile view plot. If you were looking at the F(t) data analyzed in the paper you would

have to have chosen the height in the water column to coincide with the position of the fragment and the two-minute averaging would bury the strong return on a single spike into noise level of the F(t) curve.

As I indicated in the text you can see fragment returns as a few high intensity (usually yellow to red) returns in a single image pixel corresponding, as indicated above, to a single range cell in the returns from a single transmitted pulse. We only convinced ourselves that our model was correct when we found that we could see such pixels after a detachment event but we had to look, very carefully, for them. We included an example in our earlier 2017 ASL Report listed in the references which is available on ResearchGate, Unfortunately, a few isolated yellow-red pixels just get lost in the larger display and there is much better evidence in the literature for anchor ice detachment. We could zoom in on such pixels but all you would see is an orangey spot embedded in a sea of mostly blue and green spots on a black background ……not very enlightening.

*In Line 27 – 28, the author described "Smaller concurrent reduction were apparent in the strengths of the longest range components o the saturated surface returns." But, I was not able to find this situation.*

If you look at the surfaces around 10:00, Jan. 26 the thickness of red band thins slightly and the faint blue lines above it (which correspond to later-arriving portions of the surface signal) fade.

*P. 7, L. 36 – 38: "This pattern . . .. to completely block detection of acoustic returns from water column and surface targets." Does anchor ice covering the transducer prevent return pulse only? Emitted pulse might be prevented by anchor ice?? This is just a comment.*

The attenuation of acoustic waves by ice on the transducer is independent of wave propagation directionality…i.e. similar reductions during transmissions and receptions.

*P. 8, L. 19 – 20: "Pre-transition sensible heat fluxes, . . .. change in water temperatures measured on the ADCP instrument." Was the heat loss from the river surface to the atmosphere calculated using atmospheric conditions such as the air temperature, humidity and wind speed? When the water temperature is at the freezing point, the change in the water temperature due to the heat loss becomes to be small. In addition, the water temperature can be changed by advection.*

We used the described simple formulation employed by most operational river ice models which do not apply corrections for wind speed and humidity. Given the miniscule amounts of supercooling attained , heat loss calculations were, as described in the text, assumed proportional to a product of the ice-free river surface fraction and the depression of surface air temperatures below the freezing point.

*P. 8, L. 25 – 27: What was the heat to transform from the heat loss to the atmosphere if it was not used to form ice?*

It cooled the river, I presume.

*P. 10, L. 25 – 42: The discussion in this paragraph is interesting.*

Yes, and I think it is at the core of the physical model we develop. I must admit, that when we started the analysis portion of our work, having relatively little background in river ice studies, we were surprised by the fact that the *in situ* process wasn't the default candidate for anchor ice growth. We don't understand why Pietrovich's observations appear to have been largely ignored.

*In Line 33 – 38, particularly, the author proposed good discussion of enhancing anchor ice growth under hydrographic conditions in the Peace River. I would like to see more quantitative discussion, if it is possible. Does the total volume of suspended frazil ice and anchor ice become to be consistent with modeled value of F(t)? According to section 2, the instruments were heated. Is this a factor to suppress anchor ice growth or accumulation on the riverbed?*

We try to elaborate a bit in the edited version but we have a limited amount of information to work with. We were pretty sure that, by now, a follow-up study which included underwater video recorded at and adjacent to the monitoring site would have been available. As far as we know it hasn't been done. Since we monitor anchor ice growth indirectly, all we can say and, we believe, demonstrate in the manuscript, is that the inferred thickening of the ice layer and its associated mass, are consistent with measured frazil fractional volumes and estimated rates of heat loss to the atmosphere. The applied electric heat could not have been sufficient to seriously impact upon ice growth on the riverbed. It obviously was sufficient, as intended, to seriously suppress growth of the critical transducer faces (except possibly in one channel that we did not use in the analyses). We believe the close-in blockages we did see were largely a spillover from the adjacent layer in which the ice bridged across the transducer head, possibly avoiding direct contact with the warm surface.

*P. 11, L. 10 – P. 12, L. 2: How large the spatial (horizontal) scale of anchor ice on the riverbed? Was anchor ice distributed around the riverbed with uniform thickness? Is it possible that the instruments promotes/suppresses anchor ice formation and accumulation? Is the discussion described in these paragraphs able to be applied only for the case when instruments are deployed on a riverbed?*

The consistency of the observed variations in F(t) and the correlation with the synoptic energy input variations suggest the horizontal scales were on the order of kms.  Good evidence of this is that the measured water levels vary smoothly and coherently as can be seen in the ranges of surface returns as well as in the hydrostatic data. Subaerial video  collected  by BC  Hydro  show detachment and fragmentation of meter-sized  sheets of such ice.. Sidescan sonar measurements confirmed the large extent of this ice form but, as far as we know, have not documented their degrees of uniformity or thickness other than to order of magnitude. It is hard to say how much of our methodology would be easily transferable to, for example, ice formation on vertical surface. My suspicion is that equivalent info would be accessible with moderate amounts of input from underwater photography.

*P. 14, Eq (7): Why does the heat flux depend on the air temperature only?*

Data on other factors were apparently not sufficiently available on the modelled scales.

*P. 14, L. 6 – 23: The author described the impact of river currents on the heat loss of anchor ice in P. 10, L. 28 – 38. Can the author consider this effect to evaluate the cumulative heat flux? The cumulative heat flux of 5.6 MJ/m2 calculated from Eq. (7) may not be suitable to be used as the critical value.*

We don't have enough information on the anchor ice to go into the thermodynamic balance near the riverbed. Obviously our estimate has uncertainties and probably may vary with river speed, bottom composition etc. Only studies of additional events can address this problem. As indicated, our choice was based upon being sure that our estimates of cumulative flux began at a time when the riverbed could be assumed to be ice free.

*P. 16, L. 4 – 5: "a tendency for water level . . .." This behavior was only found during Interval 3 in Fig. 5b. Did the author mention about Interval 3 only?*

We're not sure what this question means. We'll be sure that the text indicates that the observed behaviour was unique to that one study interval.

*P. 16, L. 5 – 6: "Mean air temperature . . . associated with Intervals 4 and 5." The author described the air temperature for each Interval for the first time here. I recommend to add the panels of time series of the air temperature in Figs. 2 and 5.*

This will be done.

*P. 16, L. 19 – P. 17, L. 4: 1) The author suggested than the air temperature was a key factor of distinctions between single and multi-peak frazil events. Are there other possible factors such as wind speed and current speed? I think that turbulence is needed to bring lighter and cooler water down to the riverbed. If it is right, much water with lower temperature is brought from the river surface to riverbed. In the fact, single and multi-peak frazil events occurred during higher and lower temperatures, respectively. However, the relationship between such two situation and the air temperature was not explained.*

As noted above in responding to an earlier question, we believe the physical stability of the anchor ice layer is adversely affected by increasing the rate of growth. This is not an unusual feature of crystal growth and two references are given in the manuscript which have noted evidence for similar effects in subsurface river ice.

*This manuscript indicated that multi-peaked F(t) was attribute to detached anchor ice. I propose that anchor ice can detach at least one time for "hard" freezing conditions at Ta ≥ -15 ◦C. Then, the instruments can detect the resuspended anchor ice during the end of single peak event. Did the instrument show such an event in echogram plots or F(t)? multi-peak frazil events occurred during higher and lower temperatures, respectively.  If anchor ice was formed around the riverbed and detached, the instruments detect resuspended anchor ice at several times. This scenario can explain the multi-peak F(t) when ice advection was taken into account. How do you think about it? According to Fig. 9, the height of the instrument is a factor to separate between single- and multi-peak frazil events. Does the author have some idea to express the relationship between the instrument height and the air temperature to distinguish the two situations?*

We believe this suggestion arises, as noted above in a question about Section 3.2.2, from a misunderstanding of the measurement technique. All the peaks and features in F(t) interpreted by us as measures of frazil content corresponded to periods free from beam blockage. The height of the instrument only determines the length of the time interval between frazil onset and the onset of blockage. Blockage effects only appear after we recorded the data associated with both types of frazil intervals. At worst, a resuspended fragment of anchor ice from upstream areas would only contaminate one single pulse return if it just happened be suspended at the mid-water range selected for compiling our F(t) time series. The 2-minute averaging utilized in the generating these time series would have precluded any effects on frazil volume estimates.

---

## Author Comment (AC2) · 30 Dec 2020

**Response to Reviewer 2**

**Point by Point Response**

The paper deals with the application of a SWIPS in the peace river and the interpretation of the results. The paper also compares the findings from the SWIPS with modelled data using the CRISSP1D river ice model. I think this is interesting findings, and the application of SWIPS could provide new insight into the formation and transport of frazil and anchor ice in rivers. So, I think this could be a valuable paper for ice researchers. I do think some clarifications is needed in the paper and it could also benefit from a simplification of the structure and the objectives of the work. I find some of the text quite dense and detailed and sometimes hard to follow. Four events are singled out for the SWIPS analysis, it is single peak and multipeak events, there is the data from the CRISSP1D model and there are other observations from literature mixed into the discussion in chapter 3 and 4 and also in 1 and 2: I miss a clear section of the objectives of the study as a final part of the introduction. What is the main objective? Testing of SWIPS? Determining the relation between in situ anchor ice growth and frazil? Testing the CRISSP1D simulations against SWIPS data? Please guide the reader. - There is a form of reading guide at the end now which could be improved. This promises something on CRISSP1D in section 2, which only amounts to some info on the setup.

**A improved version of this would be helpful.**

Apologies. Restructuring was badly needed in the Introduction as well as a little more elaboration on the Section to Section structure. It has now been done.

There is a focus in the abstract (line 13-15) and in the introduction (30-33) which have "anomalously" low frazil content compared to the CRISSP1D model which I understand from the text does not simulate the formation of anchor ice formation. If this is the case, I am not sure I understand this comparison and the focus on the differences. If the model does not handle anchor ice properly, I do not see why this comparison is an issue at all unless you want to convey to the model developers that they need to improve their model? Or is there a previously understanding from observations that the formation of anchor ice is not a large part of the ice formation process in the Peace river? Do the discrepancies between modelled frazil and observed frazil + anchor ice development match in some way? It could be I am misunderstanding this but in the summary it seems that in-situ growth is a surprising discovery. I thought this was a well-established principle of anchor ice development, particularly in smaller rivers and streams where large quantities of anchor ice is seen developing while the amount of suspended frazil could be quite low. There are a number of works outlining this mechanism, e.g. Turcotte et al. in several papers. On page 20 you seem to reject the principle of growth of anchor ice by capture of frazil. This might be the case on a large and deep river like Peace, but I do not think this is the case if you look at anchor ice formation in general. In shallow turbulent streams accumulation (capture) of frazil should be considered, see e.g. Stickler and Alfredsen (2009, Hydrological Processes). But it could be difficult to distinguish these processes at times, and I agree with the need to address this as outlined at the end of section 4.2.

There is a complicated history here. Our initial involvement in frazil studies primarily involved helping with the deployment of instruments which we manufacture and carrying out the processing of the

acquired data. We had, in earlier processing of data from previous BC Hydro deployments, noted serious problems with physical instrument instabilities...completely losing one instrument and incurring beam blockages in all annual programs. This necessitated adding electrical heating to the instrument package. Nevertheless, after processing the results from the 2011-2012 deployment we were told that we had missed something ... the deduced fractional volume values were very far below those required to make the CRISSP1D model simulations compatible with observed surface ice volume production rates. The latter rates required frazil contents to be about two orders of magnitude larger than indicated by our data. Our description of these results in the abstract as "anomalous" relative to a model which excluded anchor ice growth reflected the fact that such a model was considered to be fully credible at the time. This view was based upon prior simulations which appeared to justify neglecting anchor ice growth since its inclusion significantly worsened agreement with surface ice growth data and introduced interpretative inconsistencies. Unfortunately, the anchor ice growth calculations were based upon the frazil capture mechanism which was, at the time, viewed as the principal or only mechanism for anchor ice growth in rivers as large as the Peace. This view was explicit in Steve Daly's remarks in the Beltaos river ice compendium. The consensus opinion, at that time, was that in situ growth was confined to small creeks and streams where it was usually morphologically detected by numerous people (Kempema, Ettema, Alfredsen, Stickler and Turcotte, among others) in the midst of other ice forms, Consequently, in its purported absence, the CRISSP1D simulations had to produce large quantities of frazil in order to be consistent with eventual production of observed amounts of surface ice. The fact that we did not see anything close to the required frazil concentrations was, indeed, anomalous. However, after much scrambling and rechecking, we concluded we had not made a major error and suggested that the fault was in the model. Not being well-steeped in the literature, it seemed obvious to us that only an *in situ* mechanism could account for what the Peace River was doing to both our instruments and to our clients' data expectations. Nevertheless, we could not convince at least two very esteemed river specialists, that we were in touch with, that this was the case...the frazil capture mechanism was apparently too well entrenched to be displaced by measurements with a new and unfamiliar technology. Most importantly, we could not get an early version of the present manuscript published, primarily because of rather hostile receptions by 2 or 3 opposed reviewers who were obviously much more influential than an almost equal number of positive reviewers. Frustratingly, the only offered objections were very vague and centred on the SWIPS calibrations which included perfectly valid measurements on polystyrene frazil surrogates. Those objections now have been, we believe, adequately addressed and supplemented with on-ice field data in a companion manuscript also recently submitted to TC.

There are two distinct camps in contention. One of which believes that frazil capture can somehow produce the large amounts of ice needed to cover a river surface. This situation persists in spite of the fact that, largely due to unofficial circulation of our results, several studies have now been carried out which have verified the predicted presence of extensive anchor ice fields in the Peace River which are hard to explain in terms of frazil capture. Also, as you note, we reference and critique, in Section 4.2, a recent example of the persistence of this alternative point of view which does not address the critical energy balance issue. We did not intend to ignore earlier in situ studies in smaller and probably more complex bodies of water. However, the physical situation appears to be simpler in larger rivers which are, as well, more accessible to both measurements and successful modelling. Just in case, we've gone over the text to be sure it's clear we are not claiming to have "invented" *in situ* anchor ice growth,

**anchor ice is released from the bottom, is drifting anchor ice captured by the SWIPS? Can this be distinguished from frazil particles? It is indicated in the text, but do we see it on the echograms?**

Such fragments should give exceptionally strong returns. The reason you don't see such returns on the Echogram is that they would only be visible if we zoomed in on a single (or maybe two) orange (high digital count) pixels surrounded by a few green or blue (low or medium digital count) pixels on a black background. This is because the signal returns from a 4 cm deep range cell at a mid-water height in our profile correspond to the average returns received over the duration of a single pulse by backscattering from targets in an imaginary 0.5 m diameter, 4 cm deep, fluid disk. The returns thus get diluted considerably and their sources are moving at about 1.25m/s with the river flow and so would be unlikely likely to be detected on more than one pulse. Moreover, given pulse durations and repetition rates, sampling at any given water level occupies only 0.01% of a given monitoring interval. That said, we **were** able see individual "anomalously" strong returns in the form of orangey single image pixels. One example of this, on the same scale as the Echogram included in the manuscript, was provided in an ASL Report listed in the references. It was not an overly impressive display. We thought about doing a zoom to highlight an strong return pixel but decided it was not worth the effort since someone would still want us to analyze the pixel statistics to prove we weren't just looking at noise. Instead, we include a reference to above-water video data showing active surfacing of anchor ice fragments.

**What level of super cooling was observed at each event in table 1? Was this measured locally, if so how?**

The fact that we were applying heat to the instrument package precluded credible measurements of supercooling and we were limited to detecting reaching the zero degree isotherm.

**Clarify how these periods were identified (start of section 3.1). You have water temperature and discharge measured at a site 370 km upstream of the SWIPS. How representative is these regarding the location of the SWIPS, e.g., how well did the model simulate the changes in water temperature over this considerable reach?**

The choices were the easy part since only 6 of the 7 intervals corresponded to significant periods of subzero water temperatures and only 4 of these preceded consolidation at the SWIPS monitoring site. The large separation between the site and location associated with the discharge and upstream water temperature data was a problem that we had to work around. BC Hydro has been using CRISSP1D together with a similar network of data inputs to guide its flood control and power production strategies with apparent success. That level of success can judged from a Fig. in the 2017 Marko et al. ASL Report (available on Research Gate) which gives the water temperatures as measured at the SWIPS site and as modelled on the basis of atmospheric parameters and 370 km distant water temperature data. We think the results are reasonably impressive but because of the complexity of a 3 dimensional river encountering tributaries along the way, getting a model that could precisely anticipate the timing of supercooling was a bridge too far. In general, the model does pretty well for above-zero temperatures

but, as far as we can tell from our acoustic data it cannot get the timing of supercooling exactly right. Consequently, comparisons with model predictions for F(t) required the shifts and artificial tricks employed as described in our manuscript. These adjustments facilitated model/measured comparisons which were of interest to show: the common form of a modelled frazil event and its incompatibility with the observed form as well as the large differences in the expected and observed F(t) magnitudes. We will add a sentence or two just above Table 2 to account for the effectiveness of such adjustments.

**Table 2, please clarify the methods used to compute the heat fluxes.**

These fluxes were equated to the product of the heat capacity of the water column under a 1 square meter of the river surface with the measured rate of water temperature decrease immediately prior to frazil onset.

**Is figure 8 necessary? Could this just have been left for the textual description?**

No it wasn't. It can be replaced by a line of text.

**Page 10: Last paragraph is interesting – could be expanded with quantification.**

I don't think we can add much here. Except for the two intervals for which we can use brief adjustments of the upstream temperature to make the environmental inputs associated with the modelled and observed frazil contents identical, the credibility of the adjusted results arises largely from the fact the shifted and unshifted model intervals are associated with similar average air temperatures and similar time trends. Given that we now know that the model's fatal flaw is ,neglecting in situ anchor ice growth, the important aspects of the model results are their estimation of magnitudes and the incompatibility of their step function time dependences with observations.

**Page 16: Is the flow the same in the single and multipeak events?**

To within 2%.

**Some minor things: - Figure 9. Provide a time axis, I think that would enhance the readability of this figure.**

Can be done.

- Provide a proper reference to Topham and Marko (2020). It is a discussion paper in C3 TCD Interactive comment Printer-friendly version Discussion paper the Cryosphere and could be referenced as such. - Provide a complete reference for Ghobrial et al. 2020 The Cryosphere. Interactive comment on The Cryosphere Discuss., https://doi.org/10.5194/tc-2020-212, 2020

Can be done.

---

## Author Response (AR1)

**Response to Reviewers**

This document is, with a few notable exceptions, similar to my first response. I think, we've managed to accommodate most requests for improvements. We still couldn't conjure up a model water temperature input anywhere near the SWIPS site but did dig up a good Echogram of newly released anchor ice. If the uploading process works, i have attached both "marked up" and unmarked versions of the revised manuscript for reviewer convenience Thank you for your patience and thoughtful suggestions.

Response to Reviewer 1

Point by Point response

Major comments

*For single peak frazil growth intervals, the large difference in measured and modeled value of frazil ice volume was shown. As the author suggested, in-situ riverbed anchor ice growth can be a factor of the difference because the river model ignored it. Did the volume of anchor ice growth on the riverbed reach to the level to explain the gap between measured and modeled frazil ice volume with 1 order of the magnitude quantitatively? Does the model overestimate suspended frazil ice volume in the case of lack of in-situ riverbed anchor ice growth? I would like to see more discussion.*

In the absence of measurements of porosity, neither of our two methods for estimating increases in anchor ice thickness directly quantifies anchor ice volume/mass production rates. Such estimates were only derived from the differences between the sensible heat loss rate immediately prior to frazil onset (derived from the cooling rate) and the latent heat produced by measured increases in frazil fractional volumes. Given that the cooling rates were generally compatible with the measured air temperatures which were also a key input for model simulations of suspended frazil fractional volume growth, our anchor ice volume production rate essentially guarantees that it accounts for the differences between modelled and measured frazil contents. If anchor ice growth were not present and buoyant frazil capture were to be the primary source of surface ice production, an effective model would have had to simulate frazil concentrations 1 to 2 orders of magnitude above measured values. It was precisely this discrepancy that necessitated our inference of dominant *in situ* anchor ice growth.

1) *The author showed that the river ice model overestimate suspended frazil ice volume. The results and discussions were based on the field data during single peak frazil growth intervals. On the other hand, these cases are not suitable to calibrate the model because of the presence of riverbed anchor ice. Are there some frazil ice growth events without the presence of riverbed anchor ice? If the model is able to estimate suspended frazil ice volume in such cases, anchor ice growth becomes to be a great factor for the model simulation*

   As far as we can tell, the rate of frazil growth is only a small fraction of that expected from the heat losses inferred from pre-event cooling rate throughout the onset of frazil growth: i.e. we immediately see from the table that initial frazil production is much less than expected from the energy balance requirement. This tells us that river and water column ice growth is initiated simultaneously, with the anchor ice growth lagging only long enough to allow for the attachment of a small number of seed frazil crystals.

2  *The author presumed riverbed and underwater situations for single- and multi-peak frazil growth intervals. These situations are consistent with measured variations of frazil ice volume during these intervals. The author suggested that the air temperature is the key factor to induce those two situations. The multi-peak*

*frazil events were induced during the periods of cooler air temperatures. According to the discussion of section 3.2.2, accumulated anchor ice layer became thicker during higher temperature periods. However, the heat loss from the river to the atmosphere becomes larger at lower air temperatures under same wind conditions, enhancing frazil ice and anchor ice growth. I would like to see more discussion on this point.*

The reviewer's point is well taken and we believe the consistent association of multiple peak events with lower air temps and greater cooling supports our argument for this view. Nevertheless, it is based upon observations of a total of two events. However, we also were able to include supporting references to crystal sizes and the strength of ice dams under soft rather than hard cooling conditions. This interpretation is also compatible with our experience in the growth of high quality metal and semiconductor crystals in which rapid growth increases defect density and structural weakness.

*#Specific comments*

*P. 1, L. 15 – 17: "A simple physical model . . .. river ice volume and mass." I agree with your opinion. In addition to it, I would like to see quantitative discussion in the main text.*

The model is summarized for single peaks at the end of 3.2.1. Important features deduced from multipeak data are added in 3.2.2. It is extended in 3.2.2 and an integrated summary is provided in Section 4.1 (p18).

*P. 4, L. 5 – 6: "Detailed analysis were confined to four of five major supercooling events," Why did the author focus on supercooling instead of suspended frazil ice detected from echogram plots? Cooler river water is lighter than warmer water at the temperature below 4 ∘C. Hence, frazil ice possibly appeared in the water column when supercooling was not detected on the riverbed.*

We make no distinction in the text between "frazil" and "supercooling" events: viewing them as interchangeable since frazil formation required supercooling. When frazil appears, it is shows up in comparable amounts throughout the water column.

*P. 5, L. 30 – 32: "These runs utilized . . .. a hydrostatic site approximately 370 km upstream of the SWIPS instrument." Was the hydrostatic site located ~370 km upstream of the SWIPS instruments site? It seems to be too far to apply the data to input the model calculation. Does the author have some comments about it?*

Yes, 370 km was very far from the SWIPS site and, yes, that distance complicated model/measurement comparisons. In fact, as we make clear in our discussions of the individual events, it usually necessitated time shifts and other adjustments in the assumed environmental forcing to allow testing of correlations with the timing of the observed frazil events. The measurements were part of a BC Hydro's annual river monitoring program which was just one of similar annual programs carried out both prior to and after the 2011-2012 work. The levels of effort were judged to be sufficient to support development and calibration of ice models for hydroelectric and flood management purposes. Clearly the detail and precision of the model comparisons could have been improved with additional, closer, upriver measurements but we do not believe this would have qualitatively altered the obtained levels of model/measurement agreement or our key conclusions regarding the roles of frazil and anchor ice. We elaborate  briefly on the underlying robustness of such comparisons in the revised manuscript.

*P. 6, L. 3: "Five separate intervals of supercooling" How large was the level of supercooling? I recommend that the author add the level of supercooling at each interval in Table 1.*

As noted in the text, heating of the instrument package was necessary to prevent immediate blockage of the acoustic beams   and package destabilization. This precluded local measurements at accuracies sufficient for tracking levels of supercooling. The water temperature data were primarily used to detect initial supercooling and estimate the rate of cooling .

*P. 6. L. 34 – 35: "The timings and intensities of the blockages, . . .. , are summarized in Fig. 3" I recommend that the author should show the situation of acoustic blockages and the air temperature at the same Figures of time series of F(t) in Figs. 2 and 5 or echogram plots in Fig. 4. Direct comparison of the timings of acoustic blockages with time evolution of F(t) or echogram plots helps us understand what the author described.*

The only significant blockages that occur  during the portions of the F(t)  records plotted in Figs 2 and 5 occurred at the ends of the records in Fig. 2 and are noted in the captions. These blockages  introduce  the steep falloff in F(t) at the end of the displayed record. It is important to note that F(t) values in periods of partial blockage are of little use for assessing water column frazil content. Meaningful content measurements require absence of anchor ice above the transducers.

*P. 7. L. 20 – 33: 1) The author pointed out critical timings such as 08:00 Jan 26, but these are difficult to be found in Fig. 4 accurately. It might be better to show such timings in Fig. 4 using some objects like as triangles.*

We have added a 08:00 marker arrow just below the time axis. There is at least a '/+/- 0.5 hour uncertainty in establishing the first faint trace of the blockage onset. Actually, judgement on this timing also can take into account information provided by the terminating drop in  F(t) in those intervals which incur blockage.

*The author mentioned the time evolution of "close-in" returns at the lowest end of the range scale, but it is too small to understand its vertical variation. In particular, the author explained that suspended frazil ice disappeared from the echogram plots due to the acoustic blockage by anchor ice at ~10:00 - ~16:00. However, the vertical evolution of the layer close to the transducer was unclear in Fig. 4 at that timing. Additional panels to enlarge the range near the transducer and to show the timings of the acoustic blockages (as shown in Fig. 3) help us understand the situations of frazil and anchor ice growth.*

We don't think this is practical and would further complicate an already complex Figure. The changes on the display scale are very slight but discernable by the trends even if the precise points where close-in thickening begins are hard to  establish from the figure and, as indicated above, should also reflect the F(t) curve..  We do introduce an expansion of the close-in region in a later figure to illustrate important changes which occur in this layer when physically significant accumulations of blockage ice are present.

*In section 3.2.2, the author suggested that anchor ice which detached from the riverbed and moved to the river surface was detected with the acoustic instruments. Why was such detached anchor ice not detected in the case shown in Fig. 4? Did the accumulated anchor ice melt and lose the thickness?*

Actually such fragments were detectable in that interval but detectability varies from Interval to Interval and is sensitive to the utilized display parameters . In the case of Fig. 4. our interests were in showing a long stretch of activity which encompassed multiple features of the backscattered signals. We just crammed in too much data to show a feature that was confined to a single pulse return. We now have added an additional data sequence as Fig. 8 which covers a shorter time period and more clearly shows the detached fragments, As I indicated in my original response remarks and in the text of the "marked up" revised manuscript, "seeing" such ice is not easy  and,

ultimately requires looking directly at full resolution digital images. In any case, the anchor ice releases have definitely have been observed both acoustically and optically. The only way of acoustically seeing such fragments is in a ProfileView plot or an equivalent. If you were looking at the F(t) data you would have to have chosen the height in the water column to coincide with the position of the fragment and the two-minute averaging would have buried strong returns on a single pulse deep into the F(t) noise levels.

*In Line 27 – 28, the author described "Smaller concurrent reduction were apparent in the strengths of the longest range components o the saturated surface returns." But, I was not able to find this situation.*

If you look at the surfaces around 10:00, Jan. 26 the thickness of red band thins slightly and the faint blue lines above it (which correspond to later-arriving portions of the surface signal) fade.

*P. 7, L. 36 – 38: "This pattern . . .. to completely block detection of acoustic returns from water column and surface targets." Does anchor ice covering the transducer prevent return pulse only? Emitted pulse might be prevented by anchor ice?? This is just a comment.*

The attenuation of acoustic waves by ice on the transducer is independent of wave propagation directionality…i.e. similar reductions during transmissions and receptions.

*P. 8, L. 19 – 20: "Pre-transition sensible heat fluxes, . . .. change in water temperatures measured on the ADCP instrument." Was the heat loss from the river surface to the atmosphere calculated using atmospheric conditions such as the air temperature, humidity and wind speed? When the water temperature is at the freezing point, the change in the water temperature due to the heat loss becomes to be small. In addition, the water temperature can be changed by advection.*

We used the described simple formulation employed by most operational river ice models which do not apply corrections for wind speed and humidity. Given the miniscule amounts of supercooling attained, heat loss calculations were, as described in the text, assumed proportional to a product of the ice-free river surface fraction and the depression of surface air temperatures below the freezing point.

*P. 8, L. 25 – 27: What was the heat to transform from the heat loss to the atmosphere if it was not used to form ice?*

It cooled the river, I presume.

*P. 10, L. 25 – 42: The discussion in this paragraph is interesting.*

Yes, and I think it is at the core of the physical model we develop. I must admit, that when we started the analysis portion of our work, having relatively little background in river ice studies, we were surprised by the fact that the *in situ* process wasn't the default candidate for anchor ice growth. We don't understand why Pietrovich's observations appear to have been largely ignored.

*In Line 33 – 38, particularly, the author proposed good discussion of enhancing anchor ice growth under hydrographic conditions in the Peace River. I would like to see more quantitative discussion, if it is possible. Does the total volume of suspended frazil ice and anchor ice become to be consistent with modeled value of F(t)?*

*According to section 2, the instruments were heated. Is this a factor to suppress anchor ice growth or accumulation on the riverbed?*

We try to elaborate a bit in the edited version but we have a limited amount of information to work with. We were pretty sure that, by now, a follow-up study which included underwater video recorded at and adjacent to the monitoring site would have been available. As far as we know it hasn't been done. Since we monitor anchor ice growth indirectly, all we can say and, we believe, demonstrate in the manuscript, is that the inferred thickening of the ice layer and its associated mass, are consistent with measured frazil fractional volumes and estimated rates of heat loss to the atmosphere. The applied electric heat could not have been sufficient to seriously impact upon ice growth on the riverbed. It obviously was sufficient, as intended, to seriously suppress growth of the critical transducer faces (except possibly in one channel that we did not use in the analyses). We believe the close-in blockages we did see were largely a spillover from the adjacent layer in which the ice bridged across the transducer head, possibly avoiding direct contact with the warm surface.

*P. 11, L. 10 – P. 12, L. 2: How large the spatial (horizontal) scale of anchor ice on the riverbed? Was anchor ice distributed around the riverbed with uniform thickness? Is it possible that the instruments promotes/suppresses anchor ice formation and accumulation? Is the discussion described in these paragraphs able to be applied only for the case when instruments are deployed on a riverbed?*

The consistency of the observed variations in F(t) and the correlation with the synoptic energy input variations suggest the horizontal scales were on the order of kms. Good evidence of this is that the measured water levels vary smoothly and coherently as can be seen in the ranges of surface returns as well as in the hydrostatic data. Subaerial video collected by BC Hydro show detachment and fragmentation of meter-sized sheets of such ice.. Sidescan sonar measurements confirmed the large extent of this ice form but, as far as we know, have not documented their degrees of uniformity or thickness other than to order of magnitude. It is hard to say how much of our methodology would be easily transferable to, for example, ice formation on vertical surface. My suspicion is that equivalent info would be accessible with moderate amounts of input from underwater photography.

*P. 14, Eq (7): Why does the heat flux depend on the air temperature only?*

Data on other factors were apparently not sufficiently available on the modelled scales.

*P. 14, L. 6 – 23: The author described the impact of river currents on the heat loss of anchor ice in P. 10, L. 28 – 38. Can the author consider this effect to evaluate the cumulative heat flux? The cumulative heat flux of 5.6 MJ/m2 calculated from Eq. (7) may not be suitable to be used as the critical value.*

We don't have enough information on the anchor ice to go into the thermodynamic balance near the riverbed. Obviously our estimate has uncertainties and probably may vary with river speed, bottom composition etc. Only studies of additional events can address this problem. As indicated, our choice was based upon being sure that our estimates of cumulative flux began at a time when the riverbed could be assumed to be ice free.

*P. 16, L. 4 – 5: "a tendency for water level . . .." This behavior was only found during Interval 3 in Fig. 5b. Did the author mention about Interval 3 only?*

We're not sure what this question means. We've modified the text to indicate that the observed behaviour was unique to that one study interval.

*P. 16, L. 5 – 6: "Mean air temperature . . . associated with Intervals 4 and 5." The author described the air temperature for each Interval for the first time here. I recommend to add the panels of time series of the air temperature in Figs. 2 and 5.*

This was done.

*P. 16, L. 19 – P. 17, L. 4: 1) The author suggested than the air temperature was a key factor of distinctions between single and multi-peak frazil events. Are there other possible factors such as wind speed and current speed? I think that turbulence is needed to bring lighter and cooler water down to the riverbed. If it is right, much water with lower temperature is brought from the river surface to riverbed. In the fact, single and multi-peak frazil events occurred during higher and lower temperatures, respectively. However, the relationship between such two situation and the air temperature was not explained.*

As noted above in responding to an earlier question, we believe the physical stability of the anchor ice layer is adversely affected by increasing the rate of growth. This is not an unusual feature of crystal growth and two references are given in the manuscript which have noted evidence for similar effects in subsurface river ice.

*This manuscript indicated that multi-peaked F(t) was attribute to detached anchor ice. I propose that anchor ice can detach at least one time for "hard" freezing conditions at Ta ≥ -15 ◦C. Then, the instruments can detect the resuspended anchor ice during the end of single peak event. Did the instrument show such an event in echogram plots or F(t)? multi-peak frazil events occurred during higher and lower temperatures, respectively. If anchor ice was formed around the riverbed and detached, the instruments detect resuspended anchor ice at several times. This scenario can explain the multi-peak F(t) when ice advection was taken into account. How do you think about it? According to Fig. 9, the height of the instrument is a factor to separate between single- and multi-peak frazil events. Does the author have some idea to express the relationship between the instrument height and the air temperature to distinguish the two situations?*

We believe this suggestion arises, as noted above in a question about Section 3.2.2, from a misunderstanding of the measurement technique. All the peaks and features in F(t) interpreted by us as measures of frazil content corresponded to periods free from beam blockage. The height of the instrument only determines the length of the time interval between frazil onset and the onset of blockage. Blockage effects only appear after we recorded the data associated with both types of frazil intervals. At worst, a resuspended fragment of anchor ice from upstream areas would only contaminate one single pulse return if it just happened be suspended at the mid-water range selected for compiling our F(t) time series. The 2-minute averaging utilized in the generating these time series would have precluded any effects on frazil volume estimates.

Response to Reviewer 2

Point by Point response

*The paper deals with the application of a SWIPS in the peace river and the interpretation of the results. The paper also compares the findings from the SWIPS with modelled data using the CRISSP1D river ice model. I think this is interesting findings, and the application of SWIPS could provide new insight into the formation and transport of frazil and anchor ice in rivers. So, I think this could be a valuable paper for ice researchers. I do think some clarifications is needed in the paper and it could also benefit from a simplification of the structure and the objectives of the work. I find some of the text quite dense and detailed and sometimes hard to follow. Four events are singled out for the SWIPS analysis, it is single peak and multipeak events, there is the data from the CRISSP1D model and there are other observations from literature mixed into the discussion in chapter 3 and 4 and also in 1*

*and 2: I miss a clear section of the objectives of the study as a final part of the introduction. What is the main objective? Testing of SWIPS? Determining the relation between in situ anchor ice growth and frazil? Testing the CRISSP1D simulations against SWIPS data? Please guide the reader. - There is a form of reading guide at the end now which could be improved. This promises something on CRISSP1D in section 2, which only amounts to some info on the setup.*

*A improved version of this would be helpful.*

Apologies. Restructuring was badly needed in the Introduction as well as a little more elaboration on the Section by Section structure. I believe we've done that.

*There is a focus in the abstract (line 13-15) and in the introduction (30-33) which have "anomalously" low frazil content compared to the CRISSP1D model which I understand from the text does not simulate the formation of anchor ice formation. If this is the case, I am not sure I understand this comparison and the focus on the differences. If the model does not handle anchor ice properly, I do not see why this comparison is an issue at all unless you want to convey to the model developers that they need to improve their model? Or is there a previously understanding from observations that the formation of anchor ice is not a large part of the ice formation process in the Peace river? Do the discrepancies between modelled frazil and observed frazil + anchor ice development match in some way? It could be I am misunderstanding this but in the summary it seems that in-situ growth is a surprising discovery. I thought this was a well-established principle of anchor ice development, particularly in smaller rivers and streams where large quantities of C2 TCD Interactive comment Printer-friendly version Discussion paper anchor ice is seen developing while the amount of suspended frazil could be quite low. There are a number of works outlining this mechanism, e.g. Turcotte et al. in several papers. On page 20 you seem to reject the principle of growth of anchor ice by capture of frazil. This might be the case on a large and deep river like Peace, but I do not think this is the case if you look at anchor ice formation in general. In shallow turbulent streams accumulation (capture) of frazil should be considered, see e.g. Stickler and Alfredsen (2009, Hydrological Processes). But it could be difficult to distinguish these processes at times, and I agree with the need to address this as outlined at the end of section 4.2.*

There is a very complicated history here. Our initial involvement in river frazil issues was primarily intended to involve helping in the deployment of instruments we manufacture and in carrying out the processing of the acquired data. We had, in processing similar data from previous BC Hydro deployments, noted serious problems with physical instrument instability…completely losing one instrument among other inconveniences. This necessitated the addition of electrical heating to the instrument package. Nevertheless, we were told that we had missed something …the deduced fractional volume values were very far below the frazil contents required to make the CRISSP1D model simulations compatible with the observed amounts of surface ice. Specifically,The required capture rates were about two orders of magnitude higher than was indicated by our data. After much scrambling and rechecking we concluded we could not have made a major error and suggested that the fault was in the model. We did realize that numerous researchers were aware of the *in situ* anchor ice form but also recognized that prior observation were almost always in very small streams and creeks. It seemed obvious to us neophytes that only the *in situ* mechanism could account for what was happening to our instruments in the Peace River in the presence of low frazil contents. Nevertheless, I could not convince at least two very esteemed river specialist reviewers that this was the case. The ambiguity of the situation was evident in the fairly recent Beltaos' review. Most importantly, we could not get an early version of the present manuscript published, primarily because of very hostile reactions by 2 or 3 svery milarly inclined reviewers in spite of the fact that the manuscript was acceptable and reasonably favourably ppreciated by an almost equal number of other reviewers. The most specific objections centred on the SWIPS calibrations which included perfectly valid measurements on polystyrene frazil surrogates. These objections were, we believe, adequately addressed in a companion manuscript which we recently submitted to TC.

There seem to be two camps here, one of which seems believes that frazil capture can somehow produce the large amounts of ice needed to cover a river surface. As you note, in section 4.2 we reference and critique a recent example of this point of view where laboratory-derived frazil concentration estimates are used to account for photographic observations of rapid anchor ice growth in a river in terms of frazil capture. I really didn't think it was worthwhile to get further enmeshed in this controversy by including an extensive historical review, but I had no intention of ignoring the considerable contributions made by people like Kempema, Turcotte and Alfredsen. I think the situation is just simpler in larger rivers and more accessible to both measurements and successful modelling. I've gone over the text to be sure it's clear we are not claiming to have "invented" *in situ* anchor ice growth

*anchor ice is released from the bottom, is drifting anchor ice captured by the SWIPS? Can this be distinguished from frazil particles? It is indicated in the text, but do we see it on the echograms?*

Such fragments should give exceptionally strong returns. Reviewer 1 had a similar question. I believe my answer, given below in a slightly modified form, also answers your question

Actually such fragments were detectable in that Interval associated with the Fig. 4 Echogram. However, ease of detectability varies from Interval to Interval and is sensitive to the utilized display parameters . In the case of Fig. 4. our interests were in showing a long stretch of activity which encompassed multiple features of the backscattered signals. We just crammed in too much data to show a feature that was confined to a single pulse return. We now have added an additional data sequence as Fig. 8 which covers a shorter time period and more clearly shows the detached fragments, As I indicated in my original response remarks and in the text of the "marked up" revised manuscript, "seeing" such ice is not easy  and, ultimately can require looking directly at full resolution digital images. In any case, the anchor ice releases have definitely have been observed both acoustically and optically. The only way of acoustically seeing such fragments would be in a ProfileView plot or an equivalent. If you were looking for fragment signatures in the F(t) data you would have to have chosen the height in the water column to coincide with the position of the fragment and the two-minute averaging would have buried the strong returns received on single pulses into the F(t) noise levels.

*Clarify how these periods were identified (start of section 3.1). You have water temperature and discharge measured at a site 370 km upstream of the SWIPS. How representative is these regarding the location of the SWIPS, e.g., how well did the model simulate the changes in water temperature over this considerable reach?*

The choices were the easy part since only 6 of the 7 intervals corresponded to significant periods of subzero water temperatures and only 4 of these preceded consolidation at the SWIPS monitoring site. The large separation between the site and location associated with the discharge and upstream water temperature data was a problem that we had to work around. BC Hydro has been using CRISSP1D together with a similar network of data inputs to guide its flood control and power production strategies with apparent success.  That level of success can judged from a Fig. in the 2017 Marko et al. ASL Report (available on Research Gate) which gives the water temperatures as measured at the SWIPS site and as modelled on the basis of atmospheric parameters and 370 km distant water temperature data. We think the results are reasonably impressive but because of the complexity of a 3 dimensional river encountering tributaries along the way, getting a model that could precisely anticipate the timing of supercooling was a bridge too far. In general, the model does pretty well for above-zero temperatures but, as far as we can tell from our acoustic data it cannot get the timing of supercooling exactly right. Consequently, comparisons with model predictions for F(t) required the shifts and artificial tricks employed as described in our manuscript. These adjustments facilitated model/measured comparisons which were of interest to show: the common form of a modelled frazil event and its incompatibility with the observed form as well as the large differences in the expected and observed F(t) magnitudes. We have added text  just above Table 2 to account for the effectiveness of such adjustments.

*Table 2, please clarify the methods used to compute the heat fluxes.*

These fluxes were equated to the product of the heat capacity of the water column under a 1 square meter of the river surface with the measured rate of water temperature decrease immediately prior to frazil onset.

*Is figure 8 necessary? Could this just have been left for the textual description?*

No it wasn't. It has been removed and replaced by a line of text.

*Page 10: Last paragraph is interesting – could be expanded with quantification.*

Additional text added ro explain insensitivity to shifts to mismatches arising from difficulties in modelling approach to supercooling with millidegree accuracy. Except for the two intervals for which we can use brief adjustments of the upstream temperature to make the environmental inputs associated with the modelled and observed frazil contents identical, the credibility of the adjusted results arises largely from the fact the shifted and unshifted model intervals are associated with similar average air temperatures and similar time trends. Given that we now know that the model's fatal flaw is neglecting in situ anchor ice growth, the important aspects of the model results are their estimation of magnitudes and the incompatibility of their step function time dependences with observations.

*Page 16: Is the flow the same in the single and multipeak events?*

To within 2%.

*Some minor things: - Figure 9. Provide a time axis, I think that would enhance the readability of this figure.*

Done

*- Provide a proper reference to Topham and Marko (2020). It is a discussion paper in C3 TCD Interactive comment Printer-friendly version Discussion paper the Cryosphere and could be referenced as such. - Provide a complete reference for Ghobrial et al. 2020 The Cryosphere. Interactive comment on The Cryosphere Discuss., https://doi.org/10.5194/tc-2020-212, 2020*

Done

---

## Author Response (AR2)

Response to Reviewers

Reviewer #1

I'm not sure I understand the reference to the sq.km scale of the frazil growth phenomena but the Reviewer's only explicit difficulties appear to be in not clearly understanding how we identify returns from the 2 different ice species. Let me be clear: although we include long term and shorter term plots showing water levels systematically responding to frazil and postulated anchor ice growth we do not do significant interpretations of these changes because, as indicated in the text, such interpretations depend upon a number of local factors on which we have no data. Instead, we rely on the strength and character of the acoustic returns which are represented by the colour and the numbers of coloured pixels in the first and third (Figs 4 and 8) Echograms to tell us how much and what kind of ice is in the water. In most cases these returns represent scatter by several individual particles of frazil in a particular range cell. Beginning near line 25 of Page 7, I have inserted words in the text to clarify what you are looking at in Fig.4. The added text in the marked up version is given in red. In Figure 8, there are much smaller numbers of high intensity green to red orange returns appearing in the water column which are indicative of returns so strong as to only be associated with large chunks of anchor ice released from the anchor ice layer. Here, additional clarifying text has been added in the Figure caption and, again, presented red in the mark up. The only other place anchor is detectable is, as you suggested, is as it collects on or above transceiver face well after the start of the frazil interval. I assume this is clear.

Reviewer #2

I agree that I was pretty emphatic about the dominance of the in situ processes and that I did run on at greater length than I originally intended in making connections with the Ghobrial and Loewen work.

In the first case, I did make it clear that very reputable people saw evidence of a veritable stew of in situ, frazil capture and, I guess, other ways of mixing ice types…but primarily in small streams, creeks etc. Our results are representative of sizable Canadian rivers with widths measured in tens and hundreds of metres and depths running up to 10 m and more. The consistency of the results we've seen in BC Hydro's annual programs and in the University of Alberta studies of several Alberta rivers makes me pretty confident that what we saw in our data was the dominant mode of anchor ice production. I don't doubt that there might be other features of a river that could contribute similar ice and we specifically note possibilities for dependences on water depth, bottom type and other factors about which have no data. Clearly detailed measurements at a single site in one river leave lots of room for new details and even possibilities of contradiction although I don't think it will be easy to explain frazil growth intervals similar to those we report within a dominant frazil capture picture.

Our original intention was to include comments on the Ghobrial and Loewen manuscript as a sort of proof of concept whereby their observations of long period of steady anchor ice growth coincided nicely with our claim that, unless in situ anchor ice releases and moves to the river surface, frazil concentrations tend to remain at low, equilibrium levels coincident with anchor ice thickening at a relatively steady rate. In trying to make this point and relate it to the Ghobrial and Loewen results, it was impossible to ignore that their lack of access to relevant ancillary data and preference for laboratory results precluded either proving their point or coming up with a more defensible interpretation. Unfortunately, their situation was not an uncommon one in this field and we believed it to useful to make suggestions for a more comprehensive research approach.